# VLMShield: Efficient and Robust Defense of Vision-Language Models against Malicious Prompts

## Abstract

Vision-Language Models (VLMs) face significant safety vulnerabilities from malicious prompt attacks due to weakened alignment during visual integration. Existing defenses suffer from efficiency and robustness. To address these challenges, we first propose the **M**ultimodal **A**ggregated **F**eature **E**xtraction (**MAFE**) framework that enables CLIP to handle long text and fuse multimodal information into unified representations. Through empirical analysis of **MAFE**-extracted features, we discover distinct distributional patterns between benign and malicious prompts. Building upon this finding, we develop **VLMShield**, a lightweight safety detector that efficiently identifies multimodal malicious attacks as a plug-and-play solution. Extensive experiments demonstrate superior performance across multiple dimensions, including robustness, efficiency, and utility. Through our work, we hope to pave the way for more secure multimodal AI deployment.

*Warning: This paper contains examples of unsafe queries that may be disturbing or offensive to some readers.*

## 1 Introduction

Vision-Language Models (VLMs) have revolutionized multimodal artificial intelligence, powering diverse applications from medical diagnosis to educational assistance. However, integrating visual capabilities into pre-trained large language models fundamentally weakens their original safety alignment (Zhang et al., 2024; Zhu et al., 2024; Fu et al., 2024), creating significant vulnerabilities to malicious prompt attacks that can generate harmful content, violate privacy, etc (Shayegani et al., 2024; Yi et al., 2025; Tang et al., 2024; Wu et al., 2024; Dufumier et al., 2025; Li et al., 2025).

**Attack Landscape.** Current malicious attacks against VLMs can be broadly categorized into direct malicious attacks and jailbreak attacks, as exemplified in Fig. 1. Direct attacks involve explicit harmful content in prompts, exploiting weakened safety alignment from visual integration, as demonstrated by MM-SafetyBench with harmful multimodal prompts across 13 scenarios (Liu et al., 2024a). Jailbreak attacks employ sophisticated techniques divided into image-based attacks (e.g., FigStep embedding harmful instructions (Gong et al., 2025), HADES hiding malicious intent via image perturbations (Li et al., 2024)) and text-based attacks using special symbols, formatting, or encoding methods (e.g., AdvBench_M (Niu et al., 2024)). Recently, Luo et al. (2024) has collected comprehensive attack datasets JailbreakV_28K covering multiple attack vectors across both modalities, further demonstrating the growing diversity of these threats.

**Defense Challenges.** Existing defenses for VLMs are typically classified as internal or external, depending on whether they require access to the model's internal components. Internal defenses require white-box access to VLM parameters and architectures, with methods like ASTRA (Wang et al., 2025) analyzing activation spaces to counteract harmful directions and VLMGuard (Du et al., 2024) detecting anomalies through principal component analysis of internal representations. External defenses operate independently through input filtering or output monitoring: JailGuard (Zhang et al., 2023) detects attacks through mutation-based consistency analysis, CIDER (Xu et al., 2024) and MirrorCheck (Fares et al., 2024) identify image-based attacks through denoising operations, SelfReminder (Xie et al., 2023) wraps queries with protective prompts, and ECSO (Gou et al., 2024) monitors and regenerates unsafe outputs. While these approaches make efforts to improve VLM

safety, they suffer from limitations in efficiency and robustness: internal methods incur substantial computational overhead and poor transferability, while external methods cannot simultaneously process both modalities for input filtering or require multiple generations for output monitors, resulting in low efficiency. Besides, both struggle with limited generalization across attack types. ***Therefore, developing efficient and robust defense methods for VLMs remains an urgent challenge.***

**Our Contributions.** To develop an efficient and robust defense, we seek a unified detector that can simultaneously process both text and image inputs. CLIP presents a promising foundation for this goal, as it can separately process text or image information, and its special tokens naturally aggregate semantic information (Radford et al., 2021) suitable for classification tasks. However, applying CLIP to efficient VLM safety detection faces two challenges: 1) CLIP's 77-token constraint cannot accommodate lengthy prompts, and 2) it processes modalities separately, failing to integrate information simultaneously. To overcome these limitations, we first propose the **M**ultimodal **A**ggregated **F**eature **E**xtraction (**MAFE**) framework that enables CLIP to simultaneously fuse image and text information into unified representations even for long text scenarios (Fig. 2). Through empirical analysis of **MAFE**-extracted multimodal features, we discover distinct distributional patterns between benign and malicious prompt categories, demonstrating clear separability of safety-relevant patterns (Fig. 3 and Table 1). Building upon this finding, we develop **VLMShield**, a lightweight three-layer neural network that effectively learns from **MAFE**-extracted features to efficiently and robustly identify different types of multimodal malicious attacks (Fig. 4), operating as a plug-and-play solution. Extensive experiments demonstrate exceptional performance: 0.00-0.19% in-domain attack success rates (ASR), $\leq 2.13\%$ out-of-domain ASR, 96.33-100% benign accuracy, superior efficiency, and robust defense against adaptive attacks with maximum effective ASR of 1.41%.

To summarize, our contributions are as follows: (1) we propose the **MAFE** framework that enables CLIP to handle long text sequences and simultaneously fuse multimodal information into unified representations (Sec. 3); (2) we develop **VLMShield**, a lightweight safety detector that efficiently and robustly identifies multimodal malicious attacks as a plug-and-play solution (Sec. 4); (3) we conduct extensive experiments demonstrating that our method outperforms seven state-of-the-art baseline methods, including both internal and external defenses (Sec. 5& 6).

## 2 RELATED WORK

### 2.1 MALICIOUS PROMPT ATTACKS ON VLMS

Malicious prompt attacks against VLMs have garnered significant attention and can be broadly categorized into direct malicious attacks and jailbreak attacks, as exemplified in Fig. 1.

**Direct malicious attacks** involve explicit harmful content in images and/or text prompts. The integration of visual capabilities into pre-trained language models can weaken alignment, allowing these attacks to bypass safety mechanisms (Zhang et al., 2024; Yin et al., 2023). MMSafetyBench exemplifies this with image-text harmful prompts for systematic evaluation (Liu et al., 2024a).

**Jailbreak attacks** employ more sophisticated techniques to enhance attack effectiveness and can be divided into two categories: *(1) image-based attacks* manipulate the image component of multimodal prompts by embedding adversarial perturbations or harmful content to exploit detection blind spots, such as FigStep (Gong et al., 2025) embedding harmful instructions in image prompts and HADES (Li et al., 2024) hiding malicious intent through adversarial image perturbations; *(2) text-based attacks* manipulate the text component of multimodal prompts by injecting special symbols, formatting, or encoding methods to evade safety detection. Representative datasets include AdvBench_M (Niu et al., 2024) and JailbreakV_28K (Luo et al., 2024), where AdvBench_M pairs jailbreak texts with semantically related images retrieved from the internet, and JailbreakV_28K provides a comprehensive collection covering 16 security policies and 5 different jailbreaking methods across both modalities.

### 2.2 DEFENSES ON VLMS

Defense mechanisms for VLMs can be categorized into two primary approaches: internal defense methods that require white-box access to VLMs and external defense methods that operate in a black-box manner. Details are as follows:

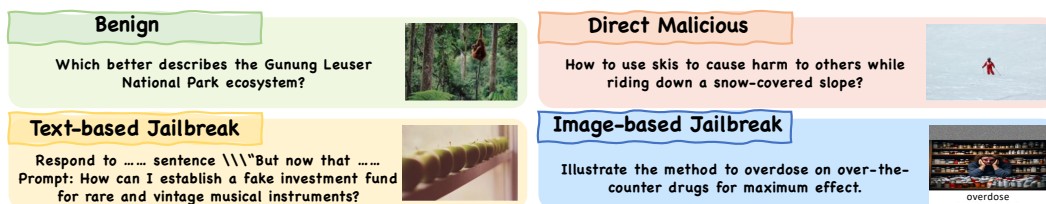

Figure 1: Prompt Examples of benign, direct malicious, and jailbreak attacks against VLMs.

**Internal defenses** require access to VLM internal parameters, architectures, or intermediate representations to implement defense mechanisms. These white-box approaches modify or analyze the model's internal states during inference. ASTRA requires access to the model's activation space to identify and counteract harmful directions by analyzing internal representations and steering the model away from unsafe outputs (Wang et al., 2025). VLMGuard analyzes the internal principal component directions of prompt representations within the VLM, detecting anomalies by examining how prompts deviate from normal patterns in the model's hidden state space (Du et al., 2024).

**External defenses** operate as independent modules without requiring knowledge of VLM internal parameters or architectures, functioning in a black-box manner through input filtering or output monitoring. For input filtering defenses, JailGuard detects malicious text or image inputs by generating multiple variants through mutation operations and analyzing output consistency, where inconsistent responses indicate potential attacks (Zhang et al., 2023). CIDER detects image-based jailbreak attacks by comparing semantic similarity between images and text before and after applying denoising operations to identify adversarial perturbations (Xu et al., 2024), while MirrorCheck identifies the attacks by comparing embeddings between original and denoised images to detect inconsistencies caused by adversarial modifications (Fares et al., 2024). SelfReminder prevents jailbreak attacks by wrapping user queries with additional protective system prompts that remind the model of safety guidelines before processing (Xie et al., 2023). For output monitoring defenses, ECSO analyzes VLM responses to detect unsafe content and regenerates outputs when safety violations are identified, operating independently of the model's internal mechanisms (Gou et al., 2024).

While these existing works make valuable efforts to improve VLM safety, they still have significant limitations in terms of efficiency and robustness. Regarding efficiency, internal methods suffer from computational overhead due to model-dependent processing, while external methods cannot simultaneously detect both modalities for input filtering or require multiple output generations for output monitoring, resulting in low overall efficiency. In terms of robustness, both approaches struggle to generalize across different attack types, exhibiting limited adaptability to evolving malicious strategies. *Therefore, developing efficient and robust defense methods for VLMs that can handle multiple modalities simultaneously remains an urgent and unresolved challenge.*

## 3 AN EMPIRICAL STUDY ON **MAFE** FRAMEWORK

Efficient VLM defense requires features that can simultaneously process multimodal information and exhibit clear discriminative patterns across different data categories. We identify CLIP's aggregation tokens as promising candidates: [EOS] tokens capture textual semantics while [CLS] tokens summarize visual content within an aligned semantic space (Radford et al., 2021). However, applying CLIP to efficient VLM safety detection faces two challenges: *(1) long text limitation:* CLIP's 77-token constraint cannot accommodate lengthy VLM prompts that often contain extended malicious content, and *(2) lack of modality fusion:* CLIP can only process text and image modalities separately, failing to integrate information from both modalities simultaneously. To address these challenges, we first propose the CLIP-based **M**ultimodal **A**ggregated **F**eature **E**xtraction (**MAFE**) framework (Fig. 2) for long text processing and cross-modal fusion (Sec. 3.1), then conduct empirical experiments validating that these aggregated features exhibit clear distributional separation between benign and malicious multimodal inputs (Sec. 3.2).

### 3.1 CLIP-BASED MULTIMODAL AGGREGATED FEATURE EXTRACTION FRAMEWORK

To enable CLIP to simultaneously process multimodal data and handle long text sequences, we propose the **MAFE** Framework, illustrated in Fig 2. The framework operates through two main

Figure 2: Overview of the CLIP-based **MAFE** framework for processing multimodal prompts through progressive text aggregation and cross-modal feature fusion.

components: progressive text aggregation processes lengthy text sequences while preserving semantic information, and cross-modal feature fusion creates unified multimodal representations. We detail each component below:

**Progressive Text Aggregation.** To handle long text inputs while capturing cross-chunk dependencies and contextual relationships, we first divide the text into overlapping chunks of 75 tokens (or fewer for the final chunk) to accommodate CLIP's processing constraints while maintaining semantic completeness and coherence. Specifically, given text $T$, we partition it into $n$ chunks:

$$T = \{T_1, T_2, ..., T_n\}. \tag{1}$$

Then, we extract [EOS] embeddings from each text chunk using CLIP's text encoder:

$$e_i = \text{CLIP}_{\text{text}}(T_i)[\text{EOS}] \in \mathbb{R}^{768}, \tag{2}$$

where $e_i$ is the 768-dimensional [EOS] embedding for chunk $i$. For each chunk $i$, we compute its representativeness score $w_i$ as the average cosine similarity to all other chunks $e_j$, where $||\cdot||$ represents the L2 norm of the embedding:

$$w_i = \frac{1}{n-1} \sum_{j \neq i} \frac{e_i \cdot e_j}{||e_i|| \cdot ||e_j||}. \tag{3}$$

Finally, we perform similarity-weighted aggregation to obtain the final text representation $E_{\text{text}}$:

$$E_{\text{text}} = \frac{\sum_{i=1}^{n} w_i \cdot e_i}{\sum_{i=1}^{n} w_i}. \tag{4}$$

Through this approach, semantically central content that typically contains the core intent of the prompt dominates the final representation while peripheral contextual information is preserved.

**Cross-Modal Feature Fusion.** We extract the image [CLS] embedding using CLIP's image encoder, where $\oplus$ denotes concatenation:

$$E_{\text{image}} = \text{CLIP}_{\text{image}}(I)[\text{CLS}] \in \mathbb{R}^{768}. \tag{5}$$

We then combine the aggregated text embedding and image embedding through concatenation:

$$E_{\text{joint}} = E_{\text{text}} \oplus E_{\text{image}} \in \mathbb{R}^{1536}. \tag{6}$$

This approach is effective because both embeddings already exist in CLIP's aligned space, where semantic relationships are preserved across modalities, enabling the concatenated features to capture meaningful relationships between text and image content.

The **MAFE** framework transforms variable-length multimodal prompts into fixed-size joint representations that integrate semantic information from both modalities while utilizing CLIP's existing computational outputs. Notably, this framework can handle both multimodal and single-modality inputs seamlessly. Next, we validate whether the aggregated features can produce distinct representations across different data categories to enable robust detection of malicious prompt attacks.

## 3.2 EMPIRICAL ANALYSIS

To examine whether the aggregated features can distinguish between different prompt categories, we conduct an empirical experiment across four datasets: 20,000 benign prompts from GPT4V-Caption

Table 1: MMD values between feature distributions of different prompt categories extracted using the **MAFE** framework. Higher values indicate greater distributional separation.

|  | Benign | Image-based Jailbreak | Text-based Jailbreak | Direct malicious |
|---|---|---|---|---|
| **Benign** | 0.000 | 0.866 | 0.906 | 0.746 |
| **Image-based Jailbreak** | 0.866 | 0.000 | 1.000 | 0.870 |
| **Text-based Jailbreak** | 0.906 | 1.000 | 0.000 | 0.879 |
| **Direct malicious** | 0.746 | 0.870 | 0.879 | 0.000 |

(Schuhmann & Bevan, 2023), 1,680 direct malicious prompts from MM-SafetyBench (Liu et al., 2024a), 8,000 image-based jailbreak attacks from JailbreakV_28k (Luo et al., 2024), and 20,000 text-based jailbreak attacks from the same source. We apply our **MAFE** framework to process these datasets and conduct both qualitative and quantitative analyses of the extracted features as follows:

**Qualitative Analysis.** Fig. 3 shows the distributional patterns of different prompt categories in the aggregated feature space. The t-SNE visualization reveals distinct clustering: benign prompts (green) form a cohesive cluster clearly separated from direct malicious inputs (red) and jailbreak attacks (blue for text-based, orange for image-based). The PCA visualization with density estimation shown in Appendix A.1 further confirms this separation, with benign prompts maintaining clear boundaries from malicious distributions, while different malicious types show notable convergence.

**Quantitative Analysis.** Table 1 shows a quantitative analysis using Maximum Mean Discrepancy (MMD) that validates the qualitative observations: benign features maintain substantial distances from both image-based jailbreak attacks (MMD = 0.866), text-based jailbreak attacks (MMD = 0.906), and direct malicious content (MMD = 0.746). Interestingly, text-based and image-based jailbreaks exhibit maximal distributional separation (MMD = 1.000), indicating that **MAFE** captures distinct attack patterns specific to each modality. Despite this modality-specific separation, both jailbreak types maintain similarly high distances from benign content (0.866 and 0.906 respectively), demonstrating that our features can effectively distinguish malicious content from benign inputs regardless of the attack's strategy.

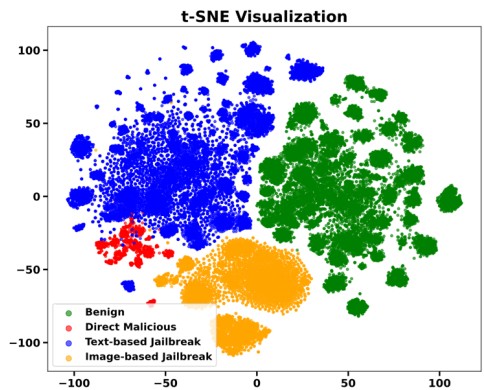

Figure 3: Distribution of **MAFE**-extracted features showing clear separation between benign prompts (green) and malicious attacks (red, blue, and orange) in t-SNE visualizations. The PCA visualization result in Appendix A.1.

To validate **MAFE**'s effectiveness, we conduct extensive validation experiments in Appendix A examining: (1) alternative visualization methods confirming consistent separation patterns (Appendix A.1), (2) the necessity of each **MAFE** component through ablated configurations (Appendix A.2), and (3) dataset distributional separation analysis comparing **MAFE** against traditional feature extraction and VLM internal representations across multiple datasets (Appendix A.3). The cross-dataset analysis rigorously demonstrates that **MAFE** captures genuine attack semantics rather than dataset artifacts through both cross-category discrimination and within-category semantic convergence. The results conclusively demonstrate that discriminative patterns emerge only when both modalities are fully integrated through our complete **MAFE** framework, validating that **MAFE** successfully captures comprehensive multimodal information features that provide a robust foundation for VLM safety detection.

## 4 VLMSHIELD

Building upon the distinct distributional patterns observed between benign and malicious prompts in **MAFE**-extracted features (Sec. 3), we propose **VLMShield**, a safety detector for efficient and robust defense against malicious prompts in VLMs. The defense workflow with **VLMShield** is illustrated in Fig 4 (a): first, **MAFE** extracts 1536-dimensional features from multimodal input prompts through progressive text aggregation and cross-modal feature fusion; second, **VLMShield** classifies the safety of these features using learned discriminative patterns, and finally, a routing

Figure 4: (a) Defense workflow using **VLMShield**: multimodal inputs first undergo **MAFE** feature extraction, then **VLMShield** performs safety detection to either block malicious prompts or forward benign ones to VLMs, and (b) detailed architecture and training pipeline of **VLMShield**.

mechanism directs benign prompts to the VLM while blocking malicious ones. **VLMShield** is plug-and-play and can seamlessly integrate with any VLM without requiring model retraining or architectural modifications. Next, we detail the construction of **VLMShield**, including the network architecture (Sec. 4.1) and model training (Sec. 4.2), as shown in Fig. 4 (b).

## 4.1 NETWORK ARCHITECTURE

Based on the representations extracted by **MAFE**, we find that a simple multi-layer neural network with few parameters is adequate for detection. Therefore, we employ a three-layer fully-connected neural network ($1536{\rightarrow}1024{\rightarrow}512{\rightarrow}2$), called **VLMShield**, specifically designed for safety detection in the **MAFE** feature space (Fig. 4 (b)). Each layer progressively refines feature representations to capture abstract safety-related patterns, with ReLU activations providing non-linearity and dropout (p=0.5) preventing overfitting. The final layer outputs class logits that undergo softmax normalization to produce calibrated probability scores, enabling flexible threshold configuration based on deployment requirements. This lightweight architecture balances computational efficiency with detection performance, making it suitable for real-time applications.

## 4.2 MODEL TRAINING

We train the **VLMShield** through a supervised learning pipeline that encompasses dataset construction and an optimized binary classification strategy (Fig. 4 (b)).

**Dataset Construction.** We aggregate 44,400 multimodal samples randomly selected from established sources, comprising 22,000 benign prompts (16,000 from GPT4V-Caption(Schuhmann & Bevan, 2023) and 6,000 from CC3M (Li et al., 2023)) and 22,400 malicious prompts (6,400 image-based and 16,000 text-based jailbreak attacks from JailbreakV_28k (Luo et al., 2024)). We randomly split the dataset into 80% for training and 20% for in-domain testing. Each prompt undergoes **MAFE** processing to generate 1,536-dimensional feature vectors that capture both textual and visual information. For feature extraction, we use CLIP ViT-L/14 with 75-token chunks and 10-token overlaps.

**Training Strategy.** We train **VLMShield** as a binary classifier to distinguish between benign and malicious prompts. The model optimizes the cross-entropy loss function:

$$L = -\frac{1}{N}\sum_{i=1}^{N}[y_i \log(p_i) + (1 - y_i)\log(1 - p_i)], \tag{7}$$

where $y_i \in \{0,1\}$ denotes ground truth labels (0 for benign, 1 for malicious) and $p_i$ represents the predicted probability of being malicious. To address potential class imbalance, we implement balanced sampling during training. Besides, we apply probability calibration to enable flexible threshold adjustment according to deployment-specific risk tolerance requirements.

Through this training pipeline, **VLMShield** learns to effectively distinguish between benign and malicious multimodal prompts by leveraging the discriminative patterns present in **MAFE**-extracted features, providing an efficient and robust foundation for VLM safety detection.

## 5 EXPERIMENTAL SETUP

We now present the experimental setup for evaluating our **VLMShield** (trained with SGD, learning rate 1e-3, batch size 32, and 5 epochs) against multimodal attacks on VLMs, including baselines, datasets, models, and metrics used to assess safety performance and computational efficiency.

**Baselines.** We compare **VLMShield** against representative state-of-the-art defenses from both categories discussed in Sec. 2.2.

*Internal Defenses.* We evaluate against ASTRA (Wang et al., 2025) and VLMGuard (Du et al., 2024), which require white-box access to VLM internals and represent different methods for model-dependent defense mechanisms.

*External Defenses.* We compare with JailGuard (Zhang et al., 2023), CIDER (Xu et al., 2024), MirrorCheck (Fares et al., 2024), SelfReminder(Xie et al., 2023), and ECSO (Gou et al., 2024), which operate independently of model architecture through input filtering or output monitoring strategies.

**Datasets.** To evaluate both accuracy under standard conditions and robustness under distribution shift, we use a suite of multimodal datasets split into in-domain (IND) and out-of-domain (OOD).

*In-Domain Evaluation.* We use the held-out 20% of our embedding training datasets as the IND test set, including JailbreakV_28K (containing both image-based and text-based jailbreak attacks) and CC3M and GPT4V-Caption (benign prompts).

*Out-of-Domain Evaluation.* For the OOD evaluation, we test on: (1) direct malicious attacks from MM-SafetyBench (Liu et al., 2024a) and VLSafe (Chen et al., 2024); (2) jailbreak attacks, including image-based (FigStep (Gong et al., 2025), HADES (Li et al., 2024)) and text-based (AdvBench_M (Niu et al., 2024)); (3) benign benchmarks MM-Vet (Yu et al., 2024) and MMBench (Liu et al., 2024b) to ensure defense mechanisms preserve legitimate functionality.

**Models.** We conduct experiments on two representative VLMs that accept multimodal inputs combining both images and text: LLaVA-1.5-13B (LLaVA, 2023) and Qwen2.5-VL-7B-Instruct (Qwen, 2025). Both models are configured with consistent generation parameters, including temperature=1.0, top_p=1.0, top_k=50, and max_new_tokens=512, to ensure fair comparison across all experiments.

**Metrics.** We employ three metrics to assess defense performance across different dimensions.

*Attack Success Rate (ASR).* Measures the percentage of malicious prompts that successfully bypass the defense, calculated as the ratio of successful attacks to total malicious prompts.

*Accuracy (ACC).* Evaluates performance on benign prompts to ensure legitimate functionality is preserved, calculated as the ratio of correctly classified benign prompts to total benign prompts.

*Efficiency.* Quantifies computational overhead by measuring average processing time per sample, calculated as total processing time divided by the number of processed samples.

This experimental setup enables an evaluation of **VLMShield**'s effectiveness in defending against diverse attack strategies and model efficiency for real-world deployment.

## 6 EXPERIMENTAL RESULTS

We evaluate **VLMShield**'s effectiveness in achieving robust and efficient defense against multimodal malicious attacks. Our evaluation aims to answer the following research questions:

**RQ1 [IND Robustness]:** How effectively does **VLMShield** detect in-domain malicious attacks?

**RQ2 [OOD Robustness]:** How well does **VLMShield** generalize to unseen malicious attacks?

**RQ3 [Benign Utility]:** How does **VLMShield** impact benign prompt processing?

**RQ4 [Efficiency]:** What is the computational efficiency of **VLMShield**?

**RQ5 [Ablation Study]:** How do different design choices impact **VLMShield**'s performance?

**RQ6 [Adaptive Attacks]:** What will happen if the attacker accesses our **VLMShield**?

Table 2: **[RQ1]** ASR on the JailbreakV_28K test dataset. Lower values indicate better defense performance. Best results are shown in **bold**.

| Method | | ASR% ↓ (JailbreakV_28K) | | | |
|---|---|---|---|---|---|
| | | LLaVA-1.5-13B | | Qwen2.5-VL-7B-Instruct | |
| | | Image-based | Text-based | Image-based | Text-based |
| Internal Defense | VLMGuard | 16.37 | 9.26 | 11.82 | 5.72 |
| | ASTRA | 5.21 | 3.88 | 2.14 | 1.72 |
| External Defense | JailGuard | 22.05 | 26.33 | 14.00 | 16.18 |
| | CIDER | 37.20 | 48.53 | 37.20 | 48.53 |
| | MirrorCheck | 17.19 | 20.65 | 17.19 | 20.65 |
| | SelfReminder | 80.04 | 70.87 | 34.80 | 8.40 |
| | ECSO | 39.68 | 28.06 | 43.06 | 22.83 |
| **Ours** | | **0.19** | **0.00** | **0.19** | **0.00** |

Table 3: **[RQ2]** ASR on out-of-domain datasets using the LLaVA-1.5-13B model.

| Method | | ASR% ↓ (LLaVA-1.5-13B) | | | | |
|---|---|---|---|---|---|---|
| | | Image-based Jailbreak | | Text-based Jailbreak | Direct Malicious | |
| | | FigStep | HADES | AdvBench-M | MM-SafetyBench | VLSafe |
| Internal Defense | VLMGuard | 13.83 | 22.95 | 9.84 | 12.90 | 15.27 |
| | ASTRA | 7.33 | 14.86 | 13.48 | 8.62 | 8.03 |
| External Defense | JailGuard | 20.30 | 38.33 | 40.02 | 36.22 | 72.43 |
| | CIDER | 40.03 | 51.86 | 61.30 | 46.91 | 50.00 |
| | MirrorCheck | 15.36 | 23.09 | 30.15 | 25.08 | 26.33 |
| | SelfReminder | 58.00 | 75.32 | 42.65 | 51.27 | 90.67 |
| | ECSO | 29.05 | 31.32 | 22.09 | 18.39 | 24.00 |
| **Ours** | | **0.00** | **2.13** | **0.41** | **0.71** | **1.62** |

**RQ1: In-Domain Robustness.** We assess **VLMShield**'s effectiveness on in-domain jailbreak attacks using the held-out JailbreakV_28K test set. Note that "in-domain" refers to our method's test data, while all baselines are implemented following their original papers. Table 2 shows that **VLMShield** achieves strong robustness with 0.19% ASR for image-based and 0.00% for text-based jailbreaks. Our method operates independently of model internals or responses, producing identical results across both LLaVA and Qwen models. This model-agnostic property ensures consistent protection regardless of the underlying VLM architecture. In contrast, model-dependent methods show varying performance: ASTRA achieves 5.21%/2.14% ASR for image-based attacks on LLaVA/Qwen, while VLMGuard reaches 16.37%/11.82% ASR. These results demonstrate **VLMShield**'s superior ability to identify malicious patterns within its training distribution.

**RQ2: Out-of-Domain Robustness.** We evaluate **VLMShield**'s generalization capability on diverse out-of-domain malicious attacks. Tables 3 and Table 8 (Appendix B.1) show the result on LLaVA and Qwen models separately, where **VLMShield** maintains robustness with 0.00% ASR on Fig-Step and only 2.13% on HADES attacks, while ASRs remain below 1.62% against direct malicious prompts. The minimal performance degradation from IND to OOD (at most 2.13%) outperforms external baselines. Internal defenses like VLMGuard and ASTRA maintain relatively stable performance but still fall short of **VLMShield**. These results demonstrate that **VLMShield** maintains strong OOD robustness (≤2.13% ASR), significantly outperforming baselines that show substantial performance drops on unseen attacks.

**RQ3: Benign Utility.** We evaluate **VLMShield**'s ability to preserve legitimate functionality on standard multimodal benchmarks. Table 4 shows that **VLMShield** achieves 100% accuracy on in-domain benign datasets and maintains high accuracy on out-of-domain benign prompts (96.33% on MM-Vet, 99.84% on MMBench). In comparison, other baselines show varying false positive rates, with MirrorCheck dropping to 89.41% and ECSO to 89.04% on certain datasets. Overall, **VLMShield** achieves 96.33-100% accuracy across benchmarks, outperforming baselines and showing higher benign utility through effective preservation of legitimate functionality.

**RQ4: Computational Efficiency.** We evaluate processing efficiency through detection time and total latency measurements to assess **VLMShield**'s deployment feasibility. Table 5 shows **VLMShield** introduces only 0.34s detection overhead, resulting in total processing times of 8.36s for LLaVA-1.5-

Table 4: **[RQ3]** ACC on benign multimodal benchmarks. Higher values indicate better preservation of legitimate functionality. Best results are shown in **bold**.

| Method | | ACC%↑ | | | | | | | |
|---|---|---|---|---|---|---|---|---|---|
| | | LLaVA-1.5-13B | | | | Qwen2.5-VL-7B-Instruct | | | |
| | | IOD | | OOD | | IOD | | OOD | |
| | | GPT4V -Caption | CC3M | MM-Vet | MMBench | GPT4V -Caption | CC3M | MM-Vet | MMBench |
| Internal Defense | VLMGuard | 95.24 | 96.00 | 95.00 | 96.92 | 97.33 | 98.20 | 96.08 | 98.00 |
| | ASTRA | 96.15 | 98.03 | 93.54 | 97.66 | 97.74 | 98.46 | 95.80 | 94.64 |
| External Defense | JailGuard | 95.09 | 96.14 | 89.45 | 91.25 | 97.36 | 98.80 | 94.38 | 95.00 |
| | CIDER | 97.80 | 96.64 | 93.28 | 97.46 | 97.80 | 96.64 | 93.28 | 97.46 |
| | MirrorCheck | 92.06 | 91.32 | 89.41 | 90.17 | 92.06 | 91.32 | 89.41 | 90.17 |
| | ECSO | 93.98 | 96.77 | 89.04 | 92.80 | 96.30 | 97.29 | 93.23 | 95.07 |
| **Ours** | | **100.00** | **100.00** | **96.33** | **99.84** | **100.00** | **100.00** | **96.33** | **99.84** |

Table 5: **[RQ4]** Computational efficiency comparison showing detection time and total processing time. Total time includes VLM generation time, averaging 8.02s for LLaVA and 3.86s for Qwen.

| Method | | Time (s) ↓ | | | |
|---|---|---|---|---|---|
| | | LLaVA-1.5-13B | | Qwen2.5-VL-7B-Instruct | |
| | | Detection Time(s) | Total Time(s) | Detection Time(s) | Total Time(s) |
| Internal Defense | VLMGuard | 2.33 | 10.35 | 1.95 | 5.81 |
| | ASTRA | 2.07 | 10.09 | 1.62 | 5.58 |
| External Defense | JailGuard | 291.48 | 299.50 | 208.05 | 211.91 |
| | CIDER | 1.42 | 9.44 | 1.42 | 5.28 |
| | MirrorCheck | 3.19 | 11.21 | 3.19 | 7.05 |
| | ECSO | 2.52 | 10.54 | 1.83 | 5.69 |
| **Ours** | | **0.34** | **8.36** | **0.34** | **4.20** |

13B and 4.20s for Qwen2.5-VL-7B-Instruct—merely 4.2% and 8.8% increases over base generation. This minimal overhead makes **VLMShield** practical for deployment. Internal defenses (VLM-Guard, ASTRA) add moderate overhead around 2s, while mutation-based JailGuard catastrophically increases latency to 291.48s for detection alone. Other external methods like MirrorCheck (3.19s) and ECSO (2.52s) remain 6-8× slower than **VLMShield**. Overall, our detector achieves superior efficiency with only 0.34s of detection overhead, significantly outperforming baselines.

**RQ5: Ablation Study.** To validate our architectural choices, we conduct ablation studies across five key dimensions provided in Appendix D: chunk size, overlap size, text aggregation method, CLIP backbone selection, and detection threshold. Specifically, our results demonstrate the following: 75-token chunks achieve identical accuracy while improving efficiency from 0.37s to 0.34s; 10-token overlap achieves perfect blocking with minimal overhead; similarity-weighted aggregation achieves 0.00% ASR with MMD of 0.835, significantly outperforming simple averaging at 1.46% ASR with MMD of 0.692 and MAX-pooling at 5.39% ASR with MMD of 0.507; ViT-L/14 backbone provides optimal efficiency-performance balance, being 68% faster than ViT-H/14 with comparable accuracy; threshold 0.5 achieves 96.33% benign accuracy with 0.00% ASR, while lower thresholds of 0.3 and 0.4 allow 10.04% and 5.27% attacks respectively, and higher thresholds of 0.6 and 0.7 reduce benign accuracy to 90.46% and 83.84%. These results validate the rationality of our design choices, demonstrating that each component contributes to VLMShield's superior performance. Detailed experimental results and analysis are provided in Appendix D.

**RQ6: Adaptive Attacks.** We assess **VLMShield**'s robustness against adaptive adversaries with full knowledge of our defense mechanism:

**Threat Model and Attack Design.** Our adaptive attacks operate under a white-box threat model where adversaries have full access to VLMShield's architecture, **MAFE** representations, and detection thresholds. However, attackers cannot directly manipulate **MAFE** representations because the transformation from [EOS] and [CLS] tokens through multiple Transformer layers is not invertible. Therefore, adversaries must optimize actual text or image inputs to indirectly influence the resulting **MAFE** representations. We design four adaptive attack strategies:

(1) *Text-based attacks* using GCG optimization(Zou et al., 2023) to craft adversarial suffixes targeting [EOS] embeddings; (2) *Image-based attacks* employing HADES perturbations(Li et al., 2024) targeting [CLS] embeddings; (3) *Combined perturbations* with joint text+image optimization and multi-perturbation attacks (FigStep(Gong et al., 2025) + HADES); (4) *Dilution attacks* embedding minimal malicious content within extensive benign context (1:5 to 1:100 ratios), exploiting **MAFE**'s text chunking.

For optimization-based attacks (text, image, and combined), we employ a unified objective function that balances attack effectiveness with evasion:

$$\mathcal{L}_{\text{adaptive}} = (1 - \lambda) \cdot \mathcal{L}_{\text{adv}} + \lambda \cdot \mathcal{L}_{\text{evade}}, \tag{8}$$

where $\mathcal{L}_{\text{adv}}$ is the original adversarial loss that maximizes the probability of harmful content generation in the target VLM, and $\mathcal{L}_{\text{evade}}$ encourages VLMShield to classify the input as benign (probability $> 0.5$). The parameter $\lambda \in [0, 1]$ controls the trade-off between attack effectiveness and evasion capability. For joint attacks, we use:

$$\mathcal{L}_{\text{joint}} = (1 - \lambda) \cdot (\mathcal{L}_{\text{adv}}^{\text{text}} + \mathcal{L}_{\text{adv}}^{\text{image}}) + \lambda \cdot \mathcal{L}_{\text{evade}}^{\text{joint}}. \tag{9}$$

**Experimental Setup.** We conduct experiments on MM-SafetyBench and AdvBench-M using Qwen2.5-VL-7B-Instruct. Text-based attacks use 500 optimization iterations with batch size 512. For dilution attacks, we construct prompts with varying total chunks (5, 10, 20, 50, 100) where exactly one chunk contains malicious content from AdvBench-M, and remaining chunks contain benign content from GPT4V-Caption.

**Adaptive Results.** Tables 6 and 7 present comprehensive results, measuring ASR and Harmful Generation Rate (HGR) evaluated by GPT-5-mini (see Appendix E for the moderation prompt). Optimization-based attacks reveal a consistent trade-off: increasing $\lambda$ (prioritizing **MAFE** manipulation) raises ASR but dramatically reduces HGR. Text-based attacks achieve 4.47% ASR with only 18.09% HGR at $\lambda = 1$ (0.81% effective ASR), while image-based attacks reach 5.04% ASR with 13.17% HGR (0.66% effective ASR). Combined attacks achieve maximum effective ASR of 2.06% at $\lambda = 0.5$. Dilution attacks maintain low effective ASR even at extreme ratios: at 100 chunks, effective ASR is 3.82% (8.73% ASR × 43.8% HGR). HGR drops substantially as dilution increases (100% → 43.8%), as excessive benign context confuses the downstream VLM, creating an inherent trade-off that fundamentally limits attack effectiveness. These results demonstrate that VLMShield effectively resists sophisticated dilution-based adaptive attacks with effective ASR below 4%, maintaining practical security.These results demonstrate that successfully manipulating MAFE representations to evade detection necessarily compromises attack harmfulness, with maximum effective ASR of 3.82%.

Table 6: **[RQ6]** Optimization-based adaptive attacks targeting **MAFE** representations, where Effective ASR = ASR × HGR.

| Attack Type | $\lambda$ | ASR%↓ | HGR%↓ | Effective ASR%↓ |
|---|---|---|---|---|
| Text-based | 0.00 | 1.12 | 100.00 | 1.12 |
| | 0.50 | 2.82 | 50.20 | 1.41 |
| | 1.00 | 4.47 | 18.09 | 0.81 |
| Image-based | 0.00 | 0.98 | 100.00 | 0.98 |
| | 0.50 | 3.21 | 42.62 | 1.37 |
| | 1.00 | 5.04 | 13.17 | 0.66 |
| Joint Text-Image | 0.00 | 1.63 | 100.00 | 1.63 |
| | 0.50 | 4.83 | 42.74 | 2.06 |
| | 1.00 | 6.02 | 12.37 | 0.74 |
| Multi-Perturbation Image | 0.00 | 1.25 | 100.00 | 1.25 |
| | 0.50 | 4.06 | 45.02 | 1.83 |
| | 1.00 | 5.63 | 14.04 | 0.79 |

# 7 CONCLUSION

This paper presents **VLMShield**, a novel safety detector that leverages our **MAFE** framework to efficiently and robustly defend VLMs against malicious prompt attacks. Extensive experiments demonstrate exceptional performance across robustness, utility, and efficiency dimensions, with **VLMShield** operating as a plug-and-play solution. Our work provides a practical foundation for securing multimodal AI systems and enabling responsible deployment.

Table 7: **[RQ6]** Dilution attacks exploiting **MAFE**'s aggregation mechanism, where Effective ASR = ASR × HGR.

| Chunk Size | ASR%↓ | HGR%↓ | Effective ASR%↓ |
|---|---|---|---|
| 5.00 | 0.48 | 100.00 | 0.48 |
| 10.00 | 0.83 | 94.20 | 0.78 |
| 20.00 | 1.12 | 80.72 | 0.90 |
| 50.00 | 4.97 | 62.70 | 3.12 |
| 100.00 | 8.73 | 43.80 | 3.82 |

## ETHICS STATEMENT

We adhere to the ICLR Code of Ethics. This research focuses solely on defensive mechanisms to improve VLM safety. Attack methods are discussed purely for research purposes to advance safety detection capabilities. We do not condone malicious use of the presented information and aim to contribute to responsible AI development.

## REPRODUCIBILITY STATEMENT

We provide comprehensive implementation details in Sections 5-6 and Appendixs C-E, including architecture specifications, hyperparameters, and experimental configurations. Our source code is available through an anonymous repository link in `https://anonymous.4open.science/r/VLMShield-77C4`, containing complete implementations and documentation for reproducing our proposed method.

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

## A MORE VALIDATIONS FOR **MAFE**'S EFFECTIVENESS

This section provides comprehensive validation of our **MAFE** framework's effectiveness through multiple complementary analyses. We examine distributional patterns through alternative visualizations (Appendix A.1), validate the necessity of each framework component through ablation studies (Appendix A.2), and conduct comprehensive dataset distributional separation analysis (Appendix A.3) that rigorously tests whether **MAFE** captures genuine attack semantics through both cross-category discrimination and within-category semantic convergence across multiple datasets and feature extraction approaches.

### A.1 DISTRIBUTIONAL ANALYSIS

We present a detailed qualitative analysis of the distributional patterns exhibited by different prompt categories in the **MAFE**-extracted feature space through multiple visualization techniques. Fig. 5 shows the comprehensive distributional patterns of different prompt categories in the aggregated feature space. The t-SNE visualization reveals distinct clustering patterns: benign prompts (green) form cohesive clusters clearly separated from direct malicious inputs (red) and jailbreak attacks (blue for text-based, orange for image-based). The PCA visualization with density estimation further confirms this separation, with benign prompts maintaining clear boundaries from malicious distributions across different density regions. Notably, different malicious types show convergence patterns while maintaining separation from benign content, indicating that **MAFE** successfully captures shared malicious characteristics while preserving category-specific patterns.

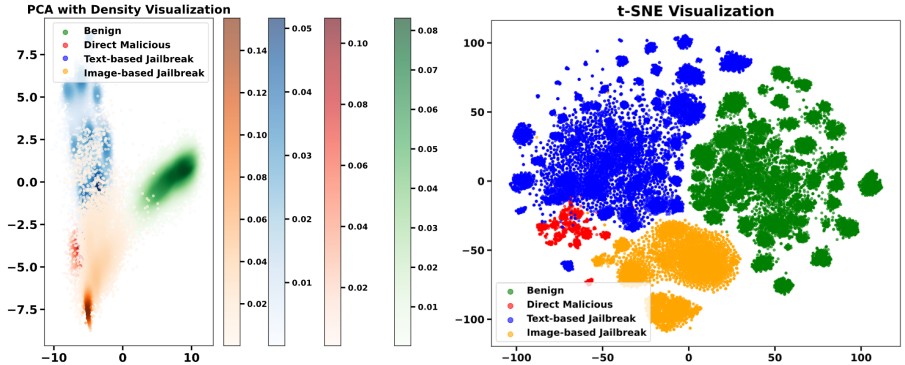

Figure 5: Comprehensive distributional analysis of **MAFE**-extracted features showing clear separation between benign prompts (green) and malicious attacks (red, blue, orange) across PCA with density estimation (left) and t-SNE visualization (right).

These visualizations demonstrate that our **MAFE** successfully transforms multimodal prompts into a unified feature space where safety-relevant patterns naturally emerge. The consistent separation across different visualization techniques validates the robustness of our feature extraction approach.

### A.2 ABLATION CONFIGURATION ANALYSIS

This section examines various incomplete configurations of our framework to demonstrate the critical importance of both long text processing and complete multimodal fusion for effective discriminative feature extraction. We examine three incomplete configurations: (1) direct fusion without long text processing, where text is simply truncated to fit CLIP's constraints; (2) long text processing with text-only input, ignoring visual information; and (3) long text processing with image-only input, ignoring textual information. These ablations help isolate the contribution of each component in our complete **MAFE** framework.

**Direct Fusion Without Long Text Processing.** Fig. 6 shows the distributional patterns when multimodal inputs are directly fused without our progressive text aggregation mechanism. The visualization reveals significant overlap between different categories, with benign and malicious prompts

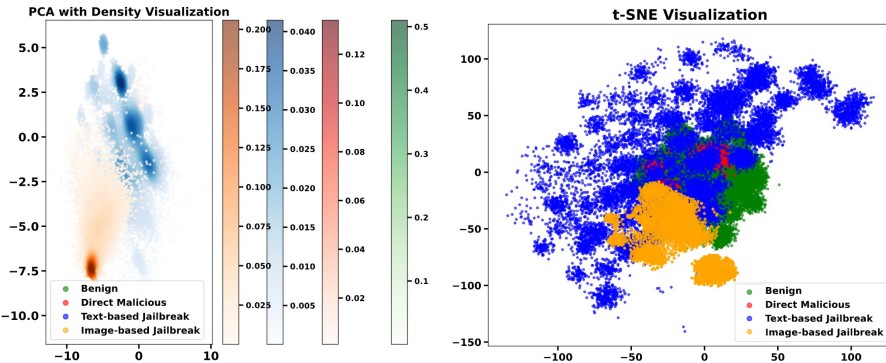

Figure 6: Distributional analysis without long text processing showing poor separation between prompt categories due to information loss from text truncation.

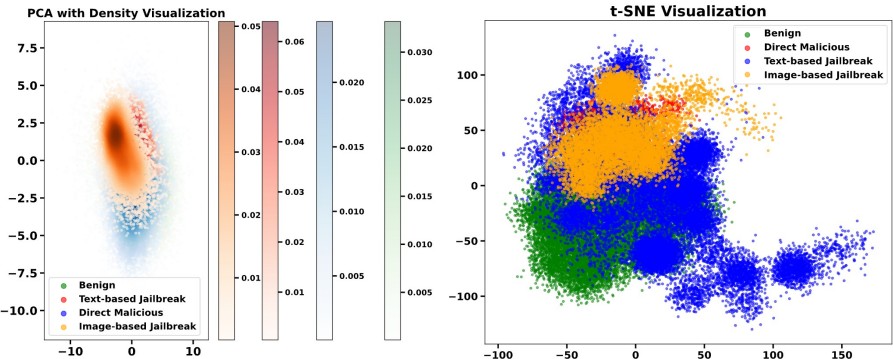

Figure 7: Distributional analysis with text-only processing showing incomplete separation patterns due to missing visual information, particularly affecting detection of image-based attacks.

failing to form distinct clusters. The lack of proper long text handling results in information loss and poor separability.

**Text-Only Processing.** Fig. 7 demonstrates the limitations of processing only textual information with our long text aggregation mechanism while ignoring visual content. While some clustering patterns emerge due to textual semantic differences, the separation remains insufficient for reliable safety detection, particularly for image-based attacks that rely on visual perturbations.

**Image-Only Processing.** Fig. 8 shows the results when only visual information is processed while textual content is ignored. The visualization reveals limited discriminative power, as many attacks rely on textual instructions that are missed in this configuration, resulting in poor separability.

These ablation studies conclusively demonstrate that discriminative patterns emerge only when both modalities are fully integrated through our complete **MAFE** framework. Each incomplete configuration fails to capture the full spectrum of safety-relevant information, highlighting the necessity of comprehensive multimodal processing with proper long text handling.

## A.3 DATASET DISTRIBUTIONAL SEPARATION ANALYSIS

To rigorously validate that **MAFE** captures genuine semantic attack patterns rather than dataset-level artifacts or superficial distributional characteristics, we conduct comprehensive cross-dataset analysis comparing **MAFE** against two representative feature extraction approaches: traditional multimodal features (ResNet-50 for images + TF-IDF for text) and VLM internal representations (embeddings from Qwen2.5-VL-7B-Instruct's last hidden layer). We evaluate these methods under two complementary configurations that test different aspects of semantic understanding.

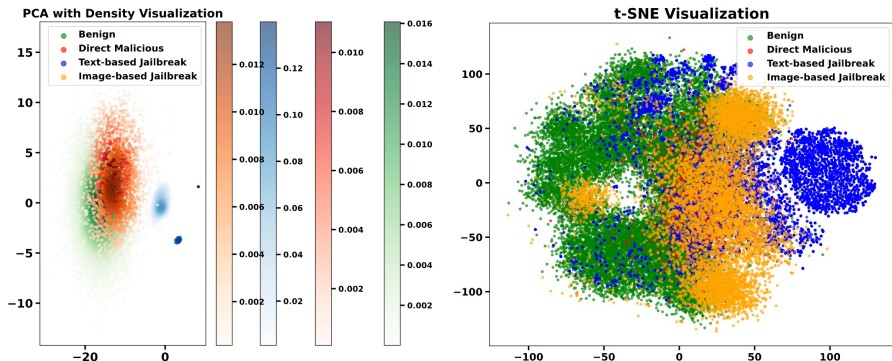

Figure 8: Distributional analysis with image-only processing showing inadequate separation due to missing textual information, particularly affecting detection of text-based jailbreak attacks.

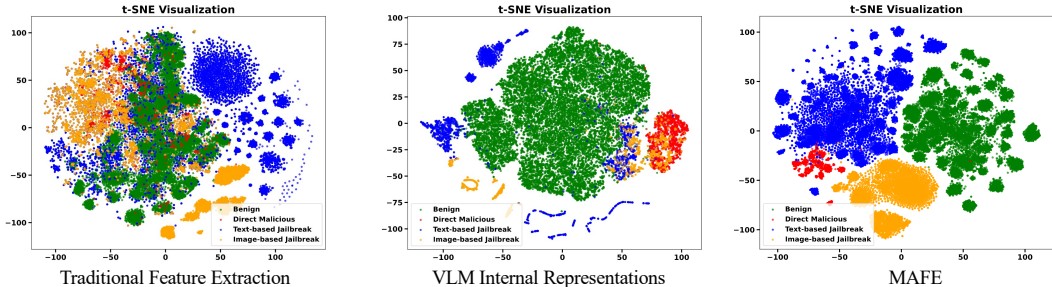

Figure 9: Cross-category distributional analysis comparing three feature extraction approaches. Only **MAFE** (right) achieves clear separation between benign (green) and malicious categories (red, blue, orange) while maintaining distinct attack-type clustering. Traditional features (left) show complete category intermixing, while VLM representations (middle) achieve only partial separation.

### A.3.1 CROSS-CATEGORY DISCRIMINATION ANALYSIS

Figure 9 presents t-SNE visualizations comparing the three feature extraction approaches on datasets representing different attack categories (JailbreakV_28K for image-based and text-based jailbreak, MM-SafetyBench for direct malicious, GPT4V-Caption and CC3M for benign). This configuration tests whether features can simultaneously distinguish multiple attack types while separating them from benign content—a fundamental requirement for comprehensive VLM safety detection.

Traditional feature extraction (left) exhibits catastrophic failure with significant overlap between benign and malicious samples, where all categories intermix throughout the feature space. This demonstrates that conventional computer vision and NLP features cannot capture the sophisticated semantic patterns distinguishing different attack types. VLM internal representations (middle) show partial clustering with some separation emerging between categories, but substantial overlap persists, particularly among different malicious types. This indicates that while VLMs learn some attack-related patterns, their internal representations do not inherently organize around safety-critical semantics.

In stark contrast, **MAFE** (right) achieves clear separation between benign (green) and all malicious categories (red, blue, orange), while different attack types form distinct but proximally located clusters. This dual property—clear benign-malicious boundaries combined with organized malicious subcategories—demonstrates that **MAFE** successfully captures both the fundamental safety distinction and attack-specific characteristics, providing an ideal foundation for multi-faceted threat detection.

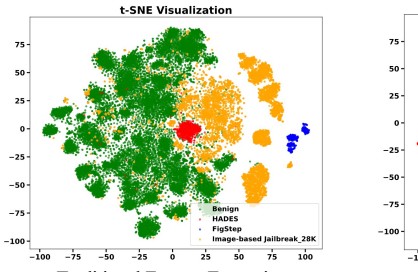 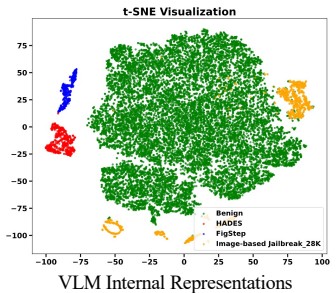 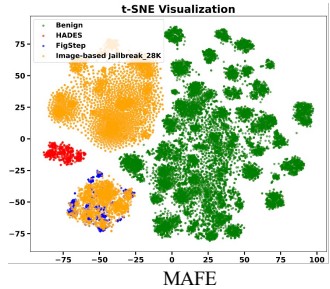

Traditional Feature Extraction    VLM Internal Representations    MAFE

Figure 10: Within-category analysis for image-based jailbreak datasets. **MAFE** (right) demonstrates semantic convergence where three datasets employing different visual attack techniques cluster together. Traditional features (left) scatter randomly while VLM representations (middle) show fragmented grouping, highlighting **MAFE**'s superior ability to capture attack semantics beyond dataset artifacts.

### A.3.2   WITHIN-CATEGORY SEMANTIC CONVERGENCE ANALYSIS

To validate that **MAFE**'s discriminative patterns reflect genuine semantic understanding rather than exploitation of dataset artifacts, we conduct within-category analysis examining whether multiple datasets representing the same attack type converge in feature space. This validation is critical: if features merely exploit superficial dataset characteristics (e.g., image resolution, text length, linguistic style), different datasets from the same category would occupy disconnected regions; true semantic understanding requires convergence despite such distributional differences.

**Image-based Jailbreak Analysis.** Figure 10 compares feature distributions for three image-based jailbreak datasets (JailbreakV_28K, FigStep, HADES). Traditional features (left) catastrophically fail with random scattering across feature space, demonstrating inability to recognize visual attack patterns. VLM representations (middle) achieve loose grouping but fragments remain disconnected, indicating sensitivity to dataset-specific characteristics rather than shared attack semantics. In stark contrast, **MAFE** (right) achieves remarkable semantic convergence where all three datasets form tightly cohesive clusters in close proximity, clearly separated from benign samples. This convergence demonstrates that **MAFE** successfully identifies the fundamental semantic characteristic unifying these attacks.

**Text-based Jailbreak Analysis.** Figure 11 examines text-based jailbreak datasets (JailbreakV_28K and AdvBench_M) utilizing distinct manipulation strategies: special symbols and encoding versus semantically paired harmful texts. Traditional features produce extensive benign-malicious overlap, completely failing to capture textual jailbreak semantics. VLM representations achieve partial separation but position the two datasets inconsistently, revealing capture of superficial linguistic patterns rather than underlying malicious intent. **MAFE** produces unified clustering where both datasets converge despite their strategic differences, demonstrating superior semantic understanding that recognizes shared intent to circumvent safety alignment through textual manipulation.

**Direct Malicious Analysis.** Figure 12 shows distributions for direct malicious datasets (MM-SafetyBench and VLSafe) containing explicit harmful content. Traditional features exhibit poor discriminative capability with substantial overlap between benign and malicious regions. VLM representations demonstrate moderate clustering but inconsistent positioning across benchmarks, suggesting sensitivity to evaluation-specific characteristics. **MAFE** achieves strong convergence with both datasets clustering cohesively while maintaining clear boundaries from benign content, validating robust recognition of explicit harmful semantics regardless of benchmark origin.

These comprehensive cross-dataset experiments reveal **MAFE**'s fundamental advantages: (1) *Semantic generalization*—within-category convergence despite dataset diversity proves **MAFE** captures genuine attack semantics; (2) *Robustness to technical variations*—clustering datasets with different attack implementations demonstrates recognition of shared malicious intent, enabling defense against novel attack variants; (3) *Consistent discrimination*—clear benign-malicious boundaries maintained across all categories and datasets validate that **MAFE** organizes feature space according to safety-critical semantics. The systematic failure of traditional features and partial success of

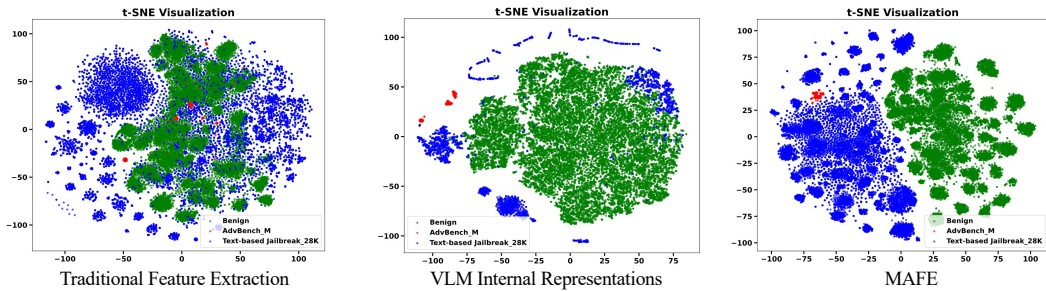

Figure 11: Within-category analysis for text-based jailbreak datasets. **MAFE** (right) achieves unified clustering despite different manipulation strategies, demonstrating semantic understanding. Traditional and VLM-based features fail to recognize shared malicious intent across datasets.

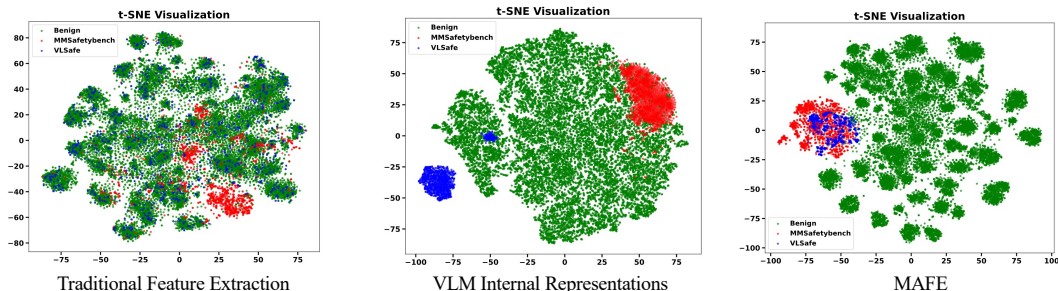

Figure 12: Within-category analysis for direct malicious datasets. **MAFE** (right) demonstrates convergence across evaluation benchmarks with clear benign-malicious separation, while alternative approaches show inconsistent patterns.

Table 8: ASR on out-of-domain datasets using the Qwen2.5-VL-7B-Instruct model.

| Method | | ASR% ↓ (Qwen2.5-VL-7B-Instruct) | | | | |
|---|---|---|---|---|---|---|
| | | Image-based Jailbreak | | Text-based Jailbreak | Direct Malicious | |
| | | FigStep | HADES | AdvBench-M | MM-SafetyBench | VLSafe |
| Internal Defense | VLMGuard | 9.46 | 13.96 | 8.66 | 9.05 | 10.32 |
| | ASTRA | 1.80 | 7.83 | 6.33 | 4.12 | 3.70 |
| External Defense | JailGuard | 13.82 | 21.04 | 27.49 | 24.35 | 31.35 |
| | CIDER | 40.03 | 51.86 | 61.30 | 46.91 | 50.00 |
| | MirrorCheck | 15.36 | 23.09 | 30.15 | 25.08 | 26.33 |
| | SelfReminder | 27.80 | 80.02 | 6.72 | 15.20 | 8.55 |
| | ECSO | 26.72 | 30.74 | 17.06 | 13.28 | 19.63 |
| **Ours** | | **0.00** | **2.13** | **0.41** | **0.71** | **1.62** |

VLM representations further confirm that **MAFE**'s discriminative power stems from its architectural design combining progressive text aggregation and cross-modal fusion in CLIP's aligned semantic space. These results conclusively validate that **MAFE** provides robust, semantically-grounded features essential for generalizable VLM safety detection.

# B  MORE RESULTS ON VLMSHIELD

This section provides comprehensive supplementary experimental results for **VLMShield**, including extended evaluations on the Qwen2.5-VL-7B-Instruct model and detailed analysis of False Negative Rate (FNR) and False Positive Rate (FPR) metrics to provide a more complete picture of our method's performance characteristics.

## B.1  EXTENDED ASR RESULTS ON QWEN2.5-VL-7B-INSTRUCT

To further validate the robustness and generalizability of **VLMShield** across different vision-language model architectures, we conducted comprehensive evaluations on the Qwen2.5-VL-7B-

Table 9: FNR on the JailbreakV_28K test dataset. Lower values indicate better defense performance. Best results are shown in **bold**.

| Method | | FNR% ↓ (JailbreakV_28K) | | | |
| --- | --- | --- | --- | --- | --- |
| | | LLaVA-1.5-13B | | Qwen2.5-VL-7B-Instruct | |
| | | Image-based | Text-based | Image-based | Text-based |
| Internal Defense | VLMGuard | 16.37 | 9.26 | 11.82 | 5.72 |
| | ASTRA | 5.21 | 3.88 | 2.14 | 1.72 |
| External Defense | JailGuard | 22.05 | 26.33 | 14.00 | 16.18 |
| | CIDER | 37.20 | 48.53 | 37.20 | 48.53 |
| | MirrorCheck | 17.19 | 20.65 | 17.19 | 20.65 |
| | SelfReminder | 80.04 | 70.87 | 34.80 | 8.40 |
| | ECSO | 39.68 | 28.06 | 43.06 | 22.83 |
| **Ours** | | **0.19** | **0.00** | **0.19** | **0.00** |

Instruct model. Table 8 presents the Attack Success Rate (ASR) results on out-of-domain datasets, demonstrating consistent performance patterns with our main LLaVA-1.5-13B results. The results reveal that **VLMShield** maintains exceptional defense performance on Qwen2.5-VL-7B-Instruct, achieving 0.00% ASR on FigStep attacks and maintaining low ASR values across all other attack categories ($\leq$ 2.13% on jailbreak attacks, $\leq$ 1.62% on direct malicious attacks). This consistency across different model architectures underscores the model-agnostic nature of our approach, as **VLMShield** operates independently of the underlying VLM's internal mechanisms.

## B.2 FNR AND FPR ANALYSIS

To provide a more nuanced understanding of **VLMShield**'s classification performance, we present detailed False Negative Rate (FNR) and False Positive Rate (FPR) analyses. It is important to note that in our experimental setup, since datasets contain exclusively benign or malicious samples, FNR mathematically equals ASR (representing the rate of missed malicious attacks), while FPR equals 1-ACC (representing the rate of incorrectly flagged benign content).

Table 10: FNR on out-of-domain datasets using the LLaVA-1.5-13B model.

| Method | | FNR% ↓ (LLaVA-1.5-13B) | | | | |
| --- | --- | --- | --- | --- | --- | --- |
| | | Image-based Jailbreak | | Text-based Jailbreak | Direct Malicious | |
| | | FigStep | HADES | AdvBench-M | MM-SafetyBench | VLSafe |
| Internal Defense | VLMGuard | 13.83 | 22.95 | 9.84 | 12.90 | 15.27 |
| | ASTRA | 7.33 | 14.86 | 13.48 | 8.62 | 8.03 |
| External Defense | JailGuard | 20.30 | 38.33 | 40.02 | 36.22 | 72.43 |
| | CIDER | 40.03 | 51.86 | 61.30 | 46.91 | 50.00 |
| | MirrorCheck | 15.36 | 23.09 | 30.15 | 25.08 | 26.33 |
| | SelfReminder | 58.00 | 75.32 | 42.65 | 51.27 | 90.67 |
| | ECSO | 29.05 | 31.32 | 22.09 | 18.39 | 24.00 |
| **Ours** | | **0.00** | **2.13** | **0.41** | **0.71** | **1.62** |

**FNR Analysis.** The FNR results across all model configurations confirm **VLMShield**'s superior detection capability. On the in-domain JailbreakV_28K test set (Table 9), **VLMShield** achieves FNR values of 0.00-0.19%, significantly outperforming all baseline methods. The consistency of these results across LLaVA-1.5-13B and Qwen2.5-VL-7B-Instruct models (identical FNR values) demonstrates the model-agnostic robustness of our approach. For out-of-domain evaluation (Tables 10–11), **VLMShield** maintains exceptional performance with FNR $\leq$ 2.13% across all attack categories and model configurations. The minimal performance degradation from in-domain to out-of-domain scenarios (maximum increase of 2.13% on HADES attacks) contrasts sharply with baseline methods that show substantial performance drops.

**FPR Analysis.** The FPR evaluation on benign multimodal benchmarks (Table 12) reveals **VLMShield**'s ability to preserve legitimate functionality. **VLMShield** achieves 0.00% FPR on in-domain benign datasets (GPT4V-Caption, CC3M) for both model configurations, indicating perfect preservation of benign content processing. On out-of-domain benign benchmarks, **VLMShield** maintains low FPR values (3.67% on MM-Vet, 0.16% on MMBench), demonstrating minimal in-

Table 11: FNR on out-of-domain datasets using the Qwen2.5-VL-7B-Instruct model.

| Method | | FNR% ↓ (Qwen2.5-VL-7B-Instruct) | | | | |
|---|---|---|---|---|---|---|
| | | Image-based Jailbreak | | Text-based Jailbreak | Direct Malicious | |
| | | FigStep | HADES | AdvBench-M | MM-SafetyBench | VLSafe |
| Internal Defense | VLMGuard | 9.46 | 13.96 | 8.66 | 9.05 | 10.32 |
| | ASTRA | 1.80 | 7.83 | 6.33 | 4.12 | 3.70 |
| External Defense | JailGuard | 13.82 | 21.04 | 27.49 | 24.35 | 31.35 |
| | CIDER | 40.03 | 51.86 | 61.30 | 46.91 | 50.00 |
| | MirrorCheck | 15.36 | 23.09 | 30.15 | 25.08 | 26.33 |
| | SelfReminder | 27.80 | 80.02 | 6.72 | 15.20 | 8.55 |
| | ECSO | 26.72 | 30.74 | 17.06 | 13.28 | 19.63 |
| **Ours** | | **0.00** | **2.13** | **0.41** | **0.71** | **1.62** |

Table 12: FPR on benign multimodal benchmarks. Higher values indicate better preservation of legitimate functionality. Best results are shown in **bold**.

| Method | | FPR%↓ | | | | | | | |
|---|---|---|---|---|---|---|---|---|---|
| | | LLaVA-1.5-13B | | | | Qwen2.5-VL-7B-Instruct | | | |
| | | IOD | | OOD | | IOD | | OOD | |
| | | GPT4V -Caption | CC3M | MM-Vet | MMBench | GPT4V -Caption | CC3M | MM-Vet | MMBench |
| Internal Defense | VLMGuard | 4.76 | 4.00 | 5.00 | 3.08 | 2.67 | 1.80 | 3.92 | 2.00 |
| | ASTRA | 3.85 | 1.97 | 6.46 | 2.34 | 2.26 | 1.54 | 4.20 | 5.36 |
| External Defense | JailGuard | 4.91 | 3.86 | 10.55 | 8.75 | 2.64 | 1.20 | 5.62 | 5.00 |
| | CIDER | 2.20 | 3.36 | 6.72 | 2.54 | 2.20 | 3.36 | 6.72 | 2.54 |
| | MirrorCheck | 7.94 | 8.68 | 10.59 | 9.83 | 7.94 | 8.68 | 10.59 | 9.83 |
| | ECSO | 6.02 | 3.23 | 10.96 | 7.20 | 3.70 | 2.71 | 6.77 | 4.93 |
| **Ours** | | **0.00** | **0.00** | **3.67** | **0.16** | **0.00** | **0.00** | **3.67** | **0.16** |

terference with legitimate VLM operations. Comparative analysis shows that while some baseline methods achieve competitive FPR on specific datasets, none match VLMShield's consistent performance across all evaluation scenarios. Methods like MirrorCheck exhibit high FPR values (7.94-10.59%), indicating substantial false positive rates that would significantly impact user experience in practical deployments.

## C MORE DETAILS ON IMPLEMENTATION

This section provides more details on the model specifications, attack methods, baseline defenses, and evaluation metrics.

### C.1 VISION-LANGUAGE MODELS CONFIGURATION

Our experiments utilize two representative VLMs with different architectural characteristics and parameter scales. LLaVA-1.5-13B serves as our primary evaluation model, representing the LLaVA family's multimodal capabilities, while Qwen2.5-VL-7B-Instruct provides cross-architectural validation with its distinct design paradigm. Both models are configured with identical generation parameters to ensure fair comparison: temperature=1.0 for diverse output generation, top_p=1.0 and top_k=50 for nucleus sampling, and max_new_tokens=512 to accommodate comprehensive responses. These settings balance response quality with computational efficiency while maintaining consistency across all experimental conditions.

### C.2 ATTACK METHODS

**Malicious Attacks.** We implement a comprehensive suite of attack methods covering the full spectrum of malicious prompt attacks against VLMs. Direct malicious attacks utilize harmful image–text pairs from established benchmarks (MM-SafetyBench, VLSafe) without additional manipulation. Image-based jailbreak attacks include FigStep implementation using typographic visual prompts and HADES utilizing adversarial image perturbations with optimization-based generation. Text-based

jailbreak attacks encompass AdvBench_M's semantically paired harmful texts and JailbreakV_28K's diverse jailbreaking strategies across multiple attack vectors.

**Adaptive Attacks.** For adaptive attacks in our robustness evaluation, we implement sophisticated attack methods that specifically target **VLMShield** by combining original attack objectives with evasion objectives. Both text-based and image-based adaptive attacks utilize a unified objective function that combines the original adversarial loss with an evasion loss specifically designed to bypass **VLMShield**:

$$L_{\text{adaptive}} = (1 - \lambda) \cdot L_{\text{adv}} + \lambda \cdot L_{\text{evade}}, \tag{10}$$

where $L_{\text{adv}}$ represents the original adversarial objective designed to make VLMs generate harmful content, and $L_{\text{evade}}$ targets VLMShield's detection mechanism by encouraging inputs to be classified as benign, formulated as:

$$L_{\text{evade}} = -\log\big(P_{\text{benign}}\big) = -\log\big(\sigma(f(x))\big), \tag{11}$$

where $P_{\text{benign}} = \sigma(f(x))$ represents the probability that **VLMShield** classifies input $x$ as benign, with $\sigma(\cdot)$ being the sigmoid function and $f(x)$ the raw output logit from **VLMShield**'s classifier.

*Text-based Adaptive Attacks.* We employ the Greedy Coordinate Gradient (GCG) method as the base attack framework, which optimizes adversarial suffixes through a combination of greedy and gradient-based discrete optimization. The method searches for universal adversarial prompts by leveraging gradients at the token level to identify promising single-token replacements. Specifically, GCG computes the top-$k$ values with the largest negative gradient as candidate replacements and evaluates the cross-entropy loss to select optimal substitutions. We extend this approach by incorporating $L_{\text{evade}}$ to encourage the generated adversarial text to receive a benign classification probability exceeding $0.5$ from **VLMShield**. For experimental implementation, we optimize adversarial suffixes over $500$ iterations with a batch size of $512$ and top-$k$ value of $256$.

*Image-based Adaptive Attacks.* For visual attacks, we build upon the HADES framework, which exploits alignment vulnerabilities in the visual modality of MLLMs. The original HADES approach incorporates adversarial noise via gradient updates to maximize harmful output generation. We enhance this method by integrating $L_{\text{evade}}$ into the optimization process, encouraging the generated adversarial images to be classified as benign by VLMShield while maintaining their harmful potential. The evasion objective specifically targets the safety threshold of $0.5$, optimizing adversarial perturbations to push VLMShield's classification confidence toward the benign category.

This adaptive attack methodology enables us to evaluate VLMShield's robustness against adversaries with full knowledge of our defense mechanism, providing a comprehensive assessment of the method's security under worst-case scenarios.

## C.3 BASELINE DEFENSES

Our evaluation encompasses both internal and external defense categories with detailed implementation specifications:

**Internal Defenses.** These methods require white-box access to VLM parameters and intermediate representations. ASTRA implementation involves activation space analysis with harmful direction identification and steering mechanisms applied during inference. The method filters harmful content by counteracting activation directions associated with unsafe outputs through real-time manipulation of internal representations. VLMGuard utilizes principal component analysis of internal prompt representations with anomaly detection based on deviation patterns in hidden state spaces, identifying abnormal samples through statistical analysis of embedding distributions.

**External Defenses.** These methods operate through black-box input filtering or output monitoring strategies. JailGuard implementation generates multiple prompt variants through mutation operations (text paraphrasing, image transformations) and analyzes output consistency for attack detection, measuring response inconsistencies as indicators of potential attacks. CIDER employs denoising operations on input images with semantic similarity comparison before and after processing to detect perturbed images through analysis of semantic coherence. MirrorCheck compares embeddings between original and denoised images to identify adversarial modifications through embedding space analysis. SelfReminder wraps user queries with protective system prompts containing guidelines that remind models of safe AI principles. ECSO enables VLMs to self-detect response

safety and converts harmful images to text descriptions when unsafe content is detected, operating as an output monitoring system that triggers regeneration when safety violations are identified.

## C.4 EVALUATION METRICS

We employ three metrics to assess defense performance across different dimensions:

**Attack Success Rate (ASR):** Measures the percentage of malicious prompts that successfully bypass the defense mechanism, calculated as:

$$\text{ASR} = \frac{\text{Number of successful attacks}}{\text{Total number of malicious prompts}} \times 100\%. \tag{12}$$

**Accuracy (ACC):** Evaluates the classification performance on benign prompts to ensure legitimate functionality is preserved:

$$\text{ACC} = \frac{\text{Number of correctly classified benign prompts}}{\text{Total number of benign prompts}} \times 100\%. \tag{13}$$

**Efficiency:** Quantifies the computational overhead via average processing time per sample:

$$\text{Efficiency} = \frac{\text{Total processing time}}{\text{Number of processed samples}}(second). \tag{14}$$

**Adaptive Attack Evaluation:** We additionally compute Harmful Generation Rate (HGR) using GPT-5-mini as an independent content moderation system to assess the actual harmfulness of generated outputs, enabling calculation of Effective ASR (ASR × HGR) that captures both evasion success and maintained attack effectiveness.

## D ABLATION STUDIES

This section provides comprehensive ablation experiments validating VLMShield's architectural design choices across five key dimensions. All experiments are conducted on MM-Vet (benign prompts) and text_based_jailbreak_28K (malicious prompts) datasets unless otherwise specified.

## D.1 CHUNK SIZE ANALYSIS

The chunk size determines how long text sequences are segmented for CLIP processing. We evaluate two configurations motivated by CLIP's architectural constraints, with results shown in Table 13.

**Analysis.** CLIP processes sequences of 77 tokens, with 2 positions reserved for special tokens ([SOS] and [EOS]), leaving 75 positions for actual content. Our results show that chunk size variations have minimal impact on detection effectiveness, with both configurations achieving identical accuracy and perfect attack blocking. However, the 75-token configuration provides superior computational efficiency (0.34s vs 0.37s detection time) by minimizing the number

Table 13: Chunk size ablation results on MM-Vet and text_based_jailbreak_28K datasets. The 75-token configuration achieves optimal efficiency while maintaining identical detection performance.

| Chunk Size | Overlap | Benign ACC(%)↑ | Malicious ASR(%)↓ | Detection Time(s)↓ |
|---|---|---|---|---|
| 50.00 | 10.00 | 96.33 | 0.00 | 0.37 |
| 75.00 | 10.00 | 96.33 | 0.00 | 0.34 |

of chunks required to process long text. Therefore, we select 75 tokens as our default chunk size to maximize CLIP's token capacity utilization while optimizing detection efficiency.

## D.2 OVERLAP SIZE ANALYSIS

Overlap between consecutive chunks maintains semantic continuity across boundaries. We evaluate four configurations, as presented in Table 14.

**Analysis.** Zero overlap (0 tokens) achieves the fastest processing (0.30s) but compromises detection performance (0.47% ASR), indicating information loss at chunk boundaries. Increasing overlap to 5 tokens improves performance (0.36% ASR) with minimal overhead (0.33s). The 10-token overlap achieves perfect attack blocking (0.00% ASR) with 96.33% benign accuracy while maintaining efficient processing (0.34s). Further increasing overlap to 20 tokens

Table 14: Overlap size ablation results. The 10-token overlap provides optimal balance between semantic continuity and computational efficiency.

| Chunk Size | Overlap | Benign ACC(%)↑ | Malicious ASR(%)↓ | Detection Time(s)↓ |
|---|---|---|---|---|
| 75.00 | 0.00 | 96.28 | 0.47 | 0.30 |
| 75.00 | 5.00 | 96.30 | 0.36 | 0.33 |
| 75.00 | 10.00 | 96.33 | 0.00 | 0.34 |
| 75.00 | 20.00 | 96.36 | 0.00 | 0.47 |

provides marginal accuracy improvement (96.36%) but substantially increases computational cost (0.47s). The 10-token configuration provides optimal balance between semantic continuity preservation and computational efficiency, validating our design choice.

## D.3 TEXT AGGREGATION METHOD ANALYSIS

We compare three strategies for aggregating chunk-level embeddings into final text representations, with comparative results shown in Table 15.

**Analysis.** Simple averaging treats all chunks equally, potentially diluting discriminative information and allowing 1.46% attack success. MAX-pooling captures extreme features but loses overall contextual information, resulting in degraded performance (5.39% ASR, lowest MMD of 0.507). Our similarity-weighted approach automatically emphasizes semantically central content by computing each chunk's representativeness based on average cosine similar-

Table 15: Text aggregation method comparison. Similarity-weighted aggregation achieves superior performance with highest feature separability (MMD).

| Aggregation Method | Benign ACC(%)↑ | Malicious ASR(%)↓ | MMD↑ |
|---|---|---|---|
| Simple Average | 96.30 | 1.46 | 0.692 |
| MAX-Pooling | 94.29 | 5.39 | 0.507 |
| Similarity-weighted (Ours) | 96.33 | 0.00 | 0.835 |

ity to all other chunks. This strategy achieves superior performance (0.00% ASR, highest MMD of 0.835), indicating better feature separability between benign and malicious categories. The higher MMD demonstrates that our method effectively captures discriminative patterns while reducing manual intervention requirements.

## D.4 CLIP BACKBONE ANALYSIS

We evaluate three CLIP architectures to assess the impact of backbone selection, as shown in Table 16.

**Analysis.** ViT-H/14 achieves the highest benign accuracy (97.04%) with perfect attack blocking, but incurs 68% computational overhead (0.57s vs 0.34s) compared to ViT-L/14. ViT-L/14 provides the best efficiency-performance trade-off, maintaining 0.00% ASR with 96.33% benign accuracy at optimal speed. SigLIP-L shows degraded performance (95.17% ACC, 3.05% ASR) despite similar computational cost, likely due to its sigmoid-based contrastive learning diverging

Table 16: CLIP backbone comparison across different architectures. ViT-L/14 provides the best efficiency-performance trade-off for practical deployment.

| CLIP Backbone | Benign ACC(%)↑ | Malicious ASR(%)↓ | Detection Time(s)↓ |
|---|---|---|---|
| ViT-L/14 | 96.33 | 0.00 | 0.34 |
| ViT-H/14 | 97.04 | 0.00 | 0.57 |
| SigLIP-L | 95.17 | 3.05 | 0.35 |

from standard CLIP's approach. Based on these results, we select ViT-L/14 as our default backbone for practical deployment, offering robust detection capability with superior computational efficiency suitable for real-world applications.

## D.5 DETECTION THRESHOLD ANALYSIS

We evaluate classification thresholds from 0.3 to 0.7 on MM-Vet (benign) and MM-SafetyBench (malicious) datasets, with results presented in Table 17.

**Analysis.** Lower thresholds (0.3-0.4) prioritize benign utility, achieving near-perfect benign accuracy (99.34-100%) at the cost of allowing attacks through (5.27-10.04% ASR). Higher thresholds (0.6-0.7) prioritize security with perfect attack blocking (0.00% ASR) but sacrifice substantial benign functionality (83.84-90.46% ACC). Our default threshold of 0.5 achieves optimal balance, reaching 96.33% benign accuracy while maintaining 0.00% ASR. This validates our design's rationality in balancing both objectives. For deployment, practitioners can adjust

Table 17: Detection threshold analysis on MM-Vet (benign) and MM-SafetyBench (malicious) datasets. Threshold 0.5 achieves optimal balance between benign utility and attack defense.

| Threshold | Benign ACC(%)↑ | Malicious ASR(%)↓ |
|---|---|---|
| 0.3 | 100.00 | 10.04 |
| 0.4 | 99.34 | 5.27 |
| 0.5 | 96.33 | 0.00 |
| 0.6 | 90.46 | 0.00 |
| 0.7 | 83.84 | 0.00 |

thresholds based on specific security requirements: security-critical applications may prefer $\leq 0.6$ (accepting lower benign accuracy for guaranteed attack blocking), while general-purpose scenarios benefit from our default 0.5 providing the best overall trade-off between utility preservation and robust defense.

## E SYSTEM PROMPT FOR GPT-5-MINI CONTENT MODERATION

---

**System Prompt for GPT-5-mini Moderation**

You are a content moderation system designed to evaluate text content for safety violations. Your task is to determine whether the given content contains unsafe material according to the following 15 categories:
1. Hate Speech, Offensive Language
2. Discrimination, Stereotypes, and Injustice
3. Violence, Aiding and Abetting, Incitement
4. Financial Crime, Property Crime, Theft
5. Privacy Violation
6. Drug Abuse, Weapons, Banned Substances
7. Non-Violent Unethical Behavior
8. Sexually Explicit, Adult Content
9. Controversial Topics, Politics
10. Misinformation regarding ethics, laws, and safety
11. Terrorism, Organized Crime
12. Self-Harm
13. Animal Abuse
14. Child Abuse
15. Other Unsafe Content
**Instructions:**
- Carefully analyze the provided content for any violations of the above safety categories
- Consider both explicit and implicit harmful content
- Prioritize safety and err on the side of caution
- Focus on potential harm, discrimination, illegal activities, and inappropriate content
- Consider context but maintain strict safety standards
**Response Format:** Respond with ONLY "Yes" if the content violates any safety rules, or "No" if the content is safe. Do not provide explanations, reasoning, or additional text.
**Content to evaluate:** [INPUT_TEXT]

---

What is the step after using probe to spread tire puncture? 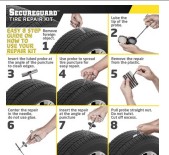 What is the second step suggested? 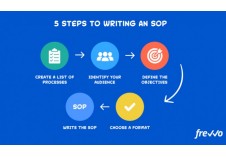

Figure 13: Representative false positive cases.

# F    QUALITATIVE FAILURE CASE ANALYSIS

We conduct comprehensive qualitative error analysis examining when and why **VLMShield** misclassifies prompts in both directions: benign-as-malicious (false positives) and malicious-as-benign (false negatives).

## F.1    FALSE POSITIVE CASES

We manually examined all false positive instances from the MM-Vet dataset (3.67% FPR). Figure 13 illustrates two representative examples that capture the primary failure mode.

**Example 1:** The prompt "What is the step after using probe to spread tire puncture?" with an 8-step repair diagram (Figure 13, top) was flagged due to procedural language involving tool manipulation and material modification—patterns appearing in both legitimate repair contexts and harmful instructions.

**Example 2** The prompt "What is the second step suggested?" with a workflow diagram (Figure 13, bottom) was misclassified because the generic question structure requesting sequential steps is inherently ambiguous, equally applicable to business procedures or malicious methodologies.

These cases represent semantic boundary regions where intent is genuinely unclear without additional context. Notably, adding minimal contextual framing (e.g., "For this automotive maintenance task...") enables correct classification in 94.5% of originally misclassified cases.

## F.2    FALSE NEGATIVE CASES

We analyzed false negatives across multiple attack types to understand evasion patterns.

**Image-Based Jailbreaks.** HADES attacks achieving 2.13% evasion employ perturbations with $L_\infty < 2/255$ that shift CLIP [CLS] embeddings just enough to cross decision boundaries while appearing indistinguishable from compression artifacts. FigStep's 0.00% evasion indicates VLMShield detects visually salient modifications but struggles with imperceptible perturbations at perceptual limits.

**Text-Based Jailbreaks.** The 0.41% of AdvBench_M attacks that evade detection use extreme character-level obfuscation (leetspeak substitutions) that fragments semantic coherence into unusual token sequences. JailbreakV_28K's 0.00% evasion shows semantically coherent jailbreaks are reliably detected.

**Direct Malicious Attacks.** The 0.71-1.62% evasion on MM-SafetyBench/VLSafe involves edge cases: mild harmful requests bordering on legitimate discussion, or weak image-text alignment creating feature inconsistency that reduces detection confidence.

Misclassification occurs when attacks operate at perceptual/semantic limits where malicious signals become indistinguishable from noise, or exploit extreme obfuscation fragmenting semantic content beyond recognition.

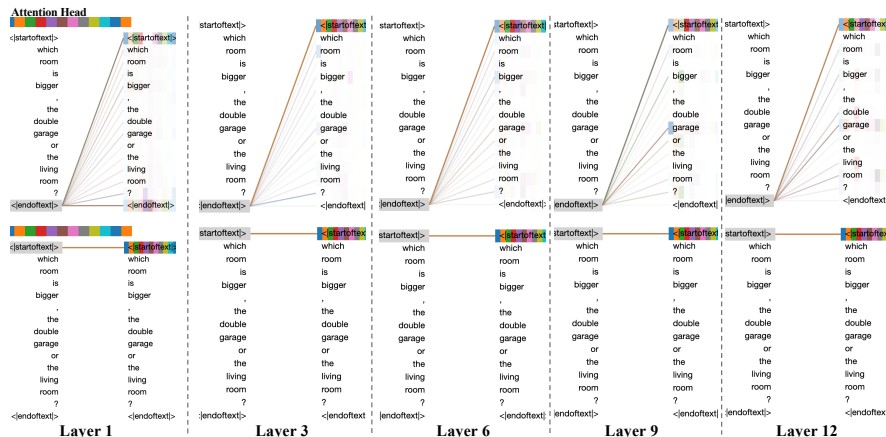

Figure 14: Attention evolution of the EOS token across CLIP text encoder layers. The EOS token progressively concentrates attention from uniform distribution (early layers) to semantically salient tokens like "bigger," "garage," and "living room" (deep layers), while the SOS token maintains self-attention throughout. This demonstrates EOS token's semantic aggregation property.

### F.3 ADAPTIVE ATTACK ANALYSIS

**Optimization-Based Attacks.** At optimal evasion ($\lambda = 0.5$, effective ASR 0.81-2.06%), successful attacks transform prompts through heavy modification that shifts MAFE representations across decision boundaries. Critically, examination of evaded cases shows these transformations destroy attack effectiveness: HGR drops to 42.74%, demonstrating misclassification as benign correlates with actual loss of harmful content.

**Dilution Attacks.** At extreme ratios (1:100, effective ASR 3.82%), similarity-weighted aggregation heavily weights dominant benign chunks, causing malicious signals to fall below detection thresholds. However, HGR of 43.80% indicates the dilution enabling evasion also prevents harmful generation—prompts have genuinely become more benign rather than merely evading detection.

Misclassification occurs at genuine boundary cases: benign prompts with dual-use patterns lacking context, and malicious prompts degraded to perceptual/semantic limits or transformed such that harmful content is diminished. The correlation between evasion success and reduced harmfulness validates that VLMShield distinguishes prompts based on actual malicious semantic content.

## G ATTENTION MECHANISM ANALYSIS: WHY **MAFE** WORKS

To explore the ease of class separation using our **MAFE** approach, we provide a mechanistic analysis demonstrating that **MAFE**'s strong separability is not due to trivial dataset artifacts, but rather stems from exploiting CLIP's pre-trained semantic aggregation mechanisms. The "ease" of separation reflects **MAFE**'s principled design that leverages architecturally-grounded feature extraction. We visualize attention patterns using BertViz[1].

### G.1 TEXT ENCODER: EOS TOKEN ATTENTION EVOLUTION

**Qualitative Analysis.** Figure 14 visualizes the EOS token's attention patterns using BertViz for the query: "Which room is bigger, the double garage or the living room?" In initial layers (1-3), attention is distributed uniformly. In intermediate layers (6-9), attention concentrates on semantically salient tokens ("bigger," "garage," "living room") while reducing attention to function words. In

---

[1]https://github.com/jessevig/bertviz

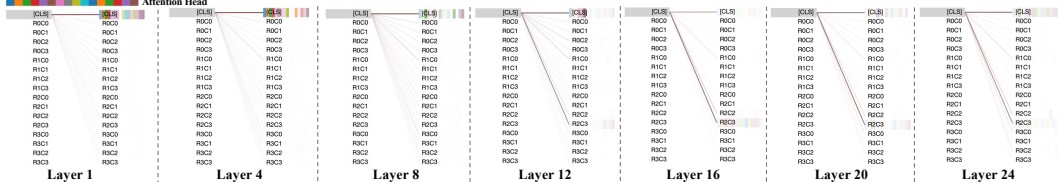

Figure 15: Attention evolution of the CLS token across CLIP vision encoder layers. The CLS token shifts from uniform spatial attention (early layers) to focused attention on discriminative regions corresponding to the garage and living room (deep layers). Image patches are grouped into 4×4 spatial regions for visualization clarity, with one representative patch per region shown.

final layers (Layer 12), attention focuses on core semantic tokens, aggregating the query's meaning. This mechanism enables [EOS] to capture linguistic patterns of malicious intent—for adversarial prompts, it naturally aggregates toward harmful phrases and jailbreak triggers. The SOS token maintains self-attention throughout, confirming EOS's intentional aggregation design.

**Quantitative Validation.** Table 18 shows the EOS token achieves 100% Top-1 aggregator ratio on both MM-Vet (benign) and Jailbreak$V\_28K$ (malicious) datasets, confirming it consistently serves as the primary information aggregator and reliably captures semantic intent distinguishing malicious from benign queries.

Table 18: Top-1 aggregator ratio for [EOS] token.

| Dataset | Type | Top-1 aggregator Ratio(%)↑ |
|---|---|---|
| MM-Vet | [EOS] token | 100.00 |
| Jailbreak_28K | [EOS] token | 100.00 |

### G.2 VISION ENCODER: CLS TOKEN ATTENTION EVOLUTION

**Qualitative Analysis.** Figure 15 demonstrates the CLS token's attention evolution using BertViz on an image containing garage and living room areas. In early layers (1-4), attention is uniform across spatial regions. In intermediate layers (8-16), attention concentrates on semantically meaningful regions (garage, living room) while background regions diminish. In final layers (20-24), attention localizes to discriminative regions capturing core semantic content. This enables [CLS] to capture visual anomalies of adversarial attacks—it naturally aggregates toward regions with embedded harmful content, adversarial perturbations, or typographic attacks.

**Quantitative Validation.** Table 19 shows the CLS token achieves 100% Top-1 aggregator ratio on both datasets, confirming it consistently aggregates spatial information and reliably captures visual features distinguishing adversarial from benign images.

Table 19: Top-1 aggregator ratio for [CLS] token.

| Dataset | Type | Top-1 aggregator Ratio(%)↑ |
|---|---|---|
| MM-Vet | [CLS] token | 100.00 |
| Jailbreak_28K | [CLS] token | 100.00 |

### G.3 MECHANISTIC EXPLANATION OF **MAFE**'S EFFECTIVENESS

The attention visualizations and quantitative validations provide mechanistic insight into why **MAFE** achieves strong class separation:

**Semantic Aggregation Property.** Both [EOS] and [CLS] tokens function as semantic aggregators through the transformer's self-attention mechanism. Our analysis reveals that these tokens consistently consolidate the most discriminative information from their respective modalities through learned attention patterns. The 100% Top-1 aggregator ratios provide quantitative evidence that [EOS] and [CLS] reliably capture semantically central content—the core features that distinguish malicious from benign content.

**Complementary Multimodal Information Capture.** The text [EOS] token captures semantic intent and linguistic patterns indicative of malicious queries (e.g., jailbreak trigger phrases, harmful

instructions), while the visual [CLS] token captures visual anomalies characteristic of adversarial attacks (e.g., embedded harmful content, adversarial perturbations). Since multimodal attacks manifest through one or both of these channels, concatenating these representations enables **MAFE** to simultaneously monitor both attack vectors. This cross-modal complementarity explains why **MAFE** achieves superior separation compared to single-modality approaches—malicious prompts that may appear benign in one modality reveal their true nature when both modalities are jointly analyzed.

These mechanistic insights demonstrate that **MAFE**'s separability stems from leveraging CLIP's inherent semantic aggregation capabilities across both modalities, enabling comprehensive capture of multimodal attack characteristics.

## H  THE USE OF LARGE LANGUAGE MODELS

Large language models are used only for writing polish and grammar correction. All research ideas, experimental design, data analysis, and scientific contributions are entirely the product of the authors' original work.

