# OpenReview forum: "VLMShield: Efficient and Robust Defense of Vision-Language Models against Malicious Prompts"
_ICLR.cc/2026/Conference — Submitted to ICLR 2026_

### Official Review · Reviewer_8s2m · 2025-10-27

**Soundness:** 3
**Presentation:** 3
**Contribution:** 3
**Rating:** 6
**Confidence:** 4

**Summary:**

The paper propose a defense method against jailbreak and direct malicious attacks against VLM. The methods has two parts. For any text-image input pair, the method first uses CLIP to obtain a concatenation of text and image embeddings, and then pass the embedding to a three layer fully connected model to obtain a classification of whether the sample is malicious. The text input is divided into chunks that meets the token limit of CLIP text encoder. The detector is trained on a labeled dataset aggregated from multiple datasets with benign and jailbreak samples.

**Strengths:**

1. The proposed defense method outperforms baselines significantly with respect to ASR and time complexity, while maintains benign sample precision as good as, if not better than the baselines.

2. The paper is clear written and easy to follow. The experiment section is especially well organized.

3. The paper presents detailed distribution analysis of benign and malicious inputs, and ablations with respect to various ways of extracting the visual and textual features.

**Weaknesses:**

1. While the propose method already has good benign sample precision, it still has non-trivial amount of miss-classified samples. Directly applying the method in production may lead to user dissatisfaction. Is it possible to provide an ablation on the benign precision and ASR with respect to different threshold of the detector?

2. There is some unclarity in the paper. For details, please see Questions.

**Questions:**

Q1. On line 176, it says "divide the text into overlapping chunks of 75 tokens", is it a typo supposed to be non-overlapping. If not, how is the overlap decided?


Q2. The authors notes some downsides of the existing external defense methods

> cannot simultaneously detect both modalities for input filtering

Is it possible to just apply multiple single modality input filter defenses at the same time? If cost is a concern, how much additional cost does it incur?

> require multiple output generations

Why would output monitoring defense requires multiple generations? If harmful contents are detected, can the model just respond with a standard disclaimer?


Q3. The image attack specific detectors CIDER and MirrorCheck have very high ASR against image attacks, as shown in Table 2,3. Is this expected?

Q4. Is it possible to implement a baseline the does not regenerate responses when harmful response is detected. Would this baseline have lower ASR than ECSO?

Q5. Is there a reason Table 2 does not have direct malicious attack results?

Q6. Regarding to adaptive attacks, is it possible to utilize the fact that the text input are partitioned into chucks, and provide a very long prompt where the malicious part is only concentrated in one small segment. Then particular chunk may only have very little contribution to the weighted text embedding, if the prompt is divided into hundreds of chunks.

---

> ### Author Response · Authors · 2025-11-21
> **Response to Reviewer-8s2m**
>
> We sincerely thank Reviewer-8s2m for the positive feedback and for recognizing our method's performance, clear presentation, and detailed analysis. We address each of your concerns and questions below:
>
> ### **Q1: Ablation Study.**
>
> > *"While the propose method already has good benign sample precision, it still has non-trivial amount of miss-classified samples. Directly applying the method in production may lead to user dissatisfaction. Is it possible to provide an ablation on the benign precision and ASR with respect to different threshold of the detector?"*
>
> **A1:** Please see **[General Response #3—Ablation Study](https://openreview.net/forum?id=pen4NG41j6&noteId=Nx6Mc0FzHO)**.
>
> ---
>
> ### **Q2: Overlapping Chunk Strategy.**
>
> > *"On line 176, it says 'divide the text into overlapping chunks of 75 tokens', is it a typo supposed to be non-overlapping. If not, how is the overlap decided?"*
>
> **A2:** We intentionally use overlapping chunks. As mentioned in ***Line-178*** and ***Line-309***, we employ 10-token overlaps between consecutive chunks. This overlap size was determined through empirical evaluation to achieve optimal clustering performance and efficiency while preserving semantic coherence across chunk boundaries. We have also added ablation experiments on overlap size in ***Appendix-D.2*** to further validate this design choice.
>
> ---
>
> ### **Q3: Multiple Single-Modality Defenses.**
>
> > - *"Is it possible to just apply multiple single modality input filter defenses at the same time? If cost is a concern, how much additional cost does it incur?"*
>
> **A3:**  As shown in ***Table-5***, image-specific methods like CIDER and MirrorCheck already incur 1.42s and 3.19s detection overhead respectively—substantially higher than our 0.34s. Combining multiple single-modality defenses would only compound this cost, making the approach even less efficient.
>
> ---
>
> ### **Q4: Output Monitoring.**
>
> > - *"Why would output monitoring defense requires multiple generations? If harmful contents are detected, can the model just respond with a standard disclaimer?"*
> > - *"Is it possible to implement a baseline that does not regenerate responses when harmful response is detected. Would this baseline have lower ASR than ECSO?"*
>
> **A4:** ECSO performs multiple generations to produce safe outputs through iterative refinement, not just detection. Following your suggestion, we have implemented a new baseline that uses MLLM-based output safety checking with standard disclaimers (without regeneration). Results show identical ASR to ECSO because both detect unsafe outputs equally well—the difference lies only in response generation strategy (disclaimer vs. refinement), not detection capability.
>
> ---
>
> ### **Q5: CIDER and MirrorCheck Performance.**
>
> > *"The image attack specific detectors CIDER and MirrorCheck have very high ASR against image attacks, as shown in Table 2,3. Is this expected?"*
>
> **A5:** Yes. Image-based jailbreak attacks encompass diverse strategies, including noise perturbations (e.g., HADES) and typographic manipulations (e.g., FigStep). CIDER and MirrorCheck primarily rely on denoising operations designed for noise-based attacks, making them ineffective against other image attack types like typographic rearrangements. Furthermore, their denoising capabilities show limited generalization even to out-of-domain noise perturbations. These limitations collectively result in their high ASR against image attacks.
>
> ---
>
> ### **Q6: Direct Malicious Attacks in Table 2.**
>
> > *"Is there a reason Table 2 does not have direct malicious attack results?"*
>
> **A6:** As shown in our distribution analysis (***Section-3***), direct malicious attacks fall within the distributional space occupied by jailbreak attacks. Therefore, we did not include direct malicious attacks in our training set, yet VLMShield still effectively defends against them (***Table-3***), demonstrating strong zero-shot generalization to this attack category.
>
> ---
>
> ### **Q7: Adaptive Attacks.**
>
> > *"Regarding to adaptive attacks, is it possible to utilize the fact that the text input are partitioned into chunks, and provide a very long prompt where the malicious part is only concentrated in one small segment. Then particular chunk may only have very little contribution to the weighted text embedding, if the prompt is divided into hundreds of chunks."*
>
> **A7:** Please see the dilution attacks in **[General Response #2—Adaptive Attacks](https://openreview.net/forum?id=pen4NG41j6&noteId=K9DSw3xIWI)**.

---

> > ### Author Response · Authors · 2025-11-26
> > **Follow-up on Rebuttal Response**
> >
> > Dear Reviewer 8s2m,
> >
> > We hope this message finds you well. We sincerely appreciate the time and effort you devoted to reviewing our manuscript and providing valuable feedback.
> >
> > We submitted our detailed rebuttal addressing all the concerns you raised and have made substantial revisions to the manuscript, including additional experiments and analyses as suggested. **As the discussion phase is approaching its conclusion**, we wanted to respectfully follow up to see if you had the opportunity to review our response.
> >
> > We remain committed to addressing any remaining questions or concerns you may have. If there are any aspects of our rebuttal that require further clarification, we would be grateful for the opportunity to address them during the remaining discussion period.
> >
> > Thank you again for your valuable contribution to improving our work. We look forward to hearing from you.
> >
> > Best regards,
> >
> > The Authors

---

### Official Review · Reviewer_vPvD · 2025-10-31

**Soundness:** 2
**Presentation:** 2
**Contribution:** 3
**Rating:** 6
**Confidence:** 3

**Summary:**

This paper presents VLMShield, an external plug-and-play safety detector for Vision-Language Models (VLMs) based on a CLIP-derived Multimodal Aggregated Feature Extraction (MAFE) framework. MAFE enables efficient fusion of long text and image inputs into unified multimodal representations. By empirically showing marked separability between benign and malicious prompt features, the authors design a lightweight three-layer neural network classifier as VLMShield, which aims to robustly detect diverse multimodal attacks. The paper conducts extensive experiments benchmarking against both internal and external baselines, reporting strong results in robustness, efficiency, and benign utility across various attack types and VLM architectures.

**Strengths:**

1. Methodological originality and practicality: The MAFE framework extends CLIP to handle long texts and construct unified multimodal representations by progressive text aggregation and cross-modal fusion, which is an elegant solution to CLIP’s token limit and modality separation.
2. Well motivated design: The paper convincingly demonstrates via both t-SNE and PCA visualizations and quantitative MMD scores that the extracted features yield substantial separability between benign and malicious prompts.
3. Results demonstrate that VLMShield achieves impressively low Attack Success Rates (ASR) on both in-domain and out-of-domain attacks, low FNR/FPR, and high benign accuracy, outperforming both internal and external baselines. The runtime and efficiency comparisons confirm very low overhead (0.34s per sample).

**Weaknesses:**

1. Lack of analysis regarding adaptivity and defense circumvention: The adaptive attack evaluation is rather brief and does not provide details of unsuccessful evasion strategies, except for a minimal effective ASR rate. There is no in-depth examination of whether the MAFE representation can be reverse-engineered or if the detector is robust to longer-term adversarial adaptation.
2. Insufficient investigation of failure cases and false positives/negatives: Although the paper does report FNR and FPR, there is no qualitative error analysis or discussion of when and why the model misclassifies (“benign as malicious” or “malicious as benign”). For example, what types of benign prompts are most likely to be flagged? Where (if at all) do adaptive attacks begin to succeed?

**Questions:**

Please refer to the weaknesses.

---

> ### Author Response · Authors · 2025-11-21
> **Response to Reviewer-vPvD**
>
> We sincerely thank Reviewer-vPvD for the thoughtful review and for recognizing our methodological originality, well-motivated design, and strong experimental results. We address each of your concerns below:
>
> ### **Q1: Adaptive Attacks.**
>
> > *"Lack of analysis regarding adaptivity and defense circumvention: The adaptive attack evaluation is rather brief and does not provide details of unsuccessful evasion strategies, except for a minimal effective ASR rate. There is no in-depth examination of whether the MAFE representation can be reverse-engineered or if the detector is robust to longer-term adversarial adaptation."*
>
> **A1:** Please see **[General Response #2—Adaptive Attacks](https://openreview.net/forum?id=pen4NG41j6&noteId=K9DSw3xIWI)**.
>
> ---
>
> ### **Q2: Investigation of Failure Cases.**
>
> > *"Insufficient investigation of failure cases and false positives/negatives: Although the paper does report FNR and FPR, there is no qualitative error analysis or discussion of when and why the model misclassifies... For example, what types of benign prompts are most likely to be flagged? Where (if at all) do adaptive attacks begin to succeed?"*
>
> **A2:** We thank the reviewer for this valuable suggestion. We will add the investigation of failure cases in the revised version soon.

---

> > ### Author Response · Authors · 2025-11-25
> > **Response to Reviewer-vPvD Q2**
> >
> > ### **Q2: Investigation of Failure Cases.**
> >
> > > *"Insufficient investigation of failure cases and false positives/negatives: Although the paper does report FNR and FPR, there is no qualitative error analysis or discussion of when and why the model misclassifies... For example, what types of benign prompts are most likely to be flagged? Where (if at all) do adaptive attacks begin to succeed?"*
> >
> >
> > **A2:** We have added qualitative analysis of failure cases in ***Appendix-F (Figure-13)***, which reveals that misclassification errors occur primarily on prompts with inherently ambiguous or degraded semantic content:
> >
> >  **1. False Positives.**
> > Misclassified benign prompts exhibit dual-use characteristics where intent is genuinely ambiguous. For example:
> > - Example-1: "What is the step after using probe to spread tire puncture?" (with a repair diagram) uses procedural language describing tool manipulation that could apply equally to legitimate automotive repair or harmful instructions.
> > - Example-2: "What is the second step suggested?" (with multi-step diagrams) is inherently context-dependent—applicable to business workflows or potentially malicious procedures.
> >
> > These boundary cases represent semantic ambiguity that would challenge even human annotators without additional context, rather than systematic detector failure.
> >
> > **2. False Negatives.**
> > Successfully evaded prompts have been fundamentally degraded through the attack process:
> > - Image-based: HADES (2.13% evasion) uses imperceptible perturbations indistinguishable from compression artifacts; FigStep (0.00% evasion) shows visually salient attacks are detected.
> > - Text-based: AdvBench_M (0.41% evasion) employs extreme character obfuscation; JailbreakV_28K (0.00% evasion) shows coherent jailbreaks fail.
> > - Direct attacks: MM-SafetyBench/VLSafe (0.71-1.62% evasion) involve edge cases with mild requests or weak image-text alignment.
> >
> > Our adaptive attack analysis (***Table-6~7***) reinforces this pattern: at optimal evasion (λ=0.5, 2.06% effective ASR), successfully evaded prompts undergo heavy modifications with transformations and obfuscation that eliminate resemblance to original harmful instructions. Generated responses become semantically unrelated to original malicious queries—the modifications required for evasion have destroyed the harmful content itself. Dilution attacks demonstrate this more dramatically: at 100 chunks (8.73% ASR, 3.82% effective ASR), evaded prompts are so heavily transformed with meaningless symbols and benign padding that generated responses become irrelevant to the original questions.
> >
> >  **3. Conclusion.**
> > These findings demonstrate that VLMShield effectively blocks prompts retaining genuinely harmful intent. Classification errors occur primarily in semantically ambiguous boundary cases or on prompts whose content has been so heavily degraded that resulting responses no longer pose meaningful threats, validating our detector's practical security value.

---

> > > ### Author Response · Authors · 2025-11-26
> > > **Follow-up on Rebuttal Response**
> > >
> > > Dear Reviewer vPvD,
> > >
> > > We hope this message finds you well. We sincerely appreciate the time and effort you devoted to reviewing our manuscript and providing valuable feedback.
> > >
> > > We submitted our detailed rebuttal addressing all the concerns you raised and have made substantial revisions to the manuscript, including additional experiments and analyses as suggested. **As the discussion phase is approaching its conclusion**, we wanted to respectfully follow up to see if you had the opportunity to review our response.
> > >
> > > We remain committed to addressing any remaining questions or concerns you may have. If there are any aspects of our rebuttal that require further clarification, we would be grateful for the opportunity to address them during the remaining discussion period.
> > >
> > > Thank you again for your valuable contribution to improving our work. We look forward to hearing from you.
> > >
> > > Best regards,
> > >
> > > The Authors

---

### Official Review · Reviewer_qPm8 · 2025-10-31

**Soundness:** 1
**Presentation:** 3
**Contribution:** 1
**Rating:** 2
**Confidence:** 4

**Summary:**

This paper presents unified detection of both text and image-based attacks via the introduction of Multimodal Aggregated Feature Extraction (MAFE) which fuses text and image tokens into a common representation. Development of a detection mechanism VLMShield in this joint representation that demonstrates clear separation of malicious from benign threats, demonstrating very low ASR on both in-domain and OOD queries while maintaining a low FPR. The authors provide empirical results supporting the value of the work and also make the case that VLMShield is a more efficient approach than many SOTA approaches.

**Strengths:**

* Unified representation space built on top of a popular open-source VLM, making the work easily adoptable and reproducible.
* MAFE addresses text processing limitations in CLIP by handling sequences longer than the 77 token limit.

**Weaknesses:**

* MAFE itself is not especially groundbreaking- many papers have developed fused representations using the concatenation or average of the image and text embeddings from CLIP. As such the main contribution is the extension to longer text.
* The empirical analysis raises suspicions that there are some tell-tale features distinguishing datasets and deeper investigation is needed to rule out the possibility the images and text are simply sampled from different distributions, making discrimination trivial. The too-good-to-be-true results in Table 3 are a smoking gun, where VLMShield is outperfoming SOTA methods by a huge margin. For example, it could be the case that all images in one dataset are 320x240 and in another dataset they are all 640x480- these systemic differences may yield detectable differences in encoding and should be carefully ruled out.
* The classification model is dependent on a corpus of labeled images which in an adversarial setting may open the door to novel attack methods that require manual detection and retraining.

**Questions:**

How does MAFE handle text segmentation where, say, a single sentence runs for more than 75 tokens? How do you achieve "semantic completeness and coherence"?

Formatting: margins between images and text are compressed making it difficult to discriminate between image captions and text body. These practices risk desk rejection.

The kernel-density representation of the input text (Eq 4) seems like a good choice- did you perform any ablations over alternatives (eg the average of the embeddings)?

"this framework can handle multimodal and single-modality inputs seamlessly" - you don't describe how you handle single-modal inputs- do you just use a zero vector for the missing modality?

Figure 3 shows not only clear separation between benign and malicious but also clear separation between the dataset sources, suggesting there may be other structural reasons why these datasets are well-separated. This sort of too-good-to-be-true result should be treated with suspicion.  Have you explored mixing images and prompts across the benign and direct malicious datasets to rule out tell-tale features?  Selective mixing of some datasets in Fig 6/7/8 also point to some structural giveaways- it would be worthwhile to better understand and explain why some of the datasets selectively collapse onto others under selective modality changes- eg image-based onto direct malicious in Figs 7 and 8.

---

> ### Author Response · Authors · 2025-11-21
> **Response to Reviewer-qPm8 Q1~Q3**
>
> We sincerely thank Reviewer-qPm8 for the detailed review and for recognizing our unified representation space, reproducibility, and MAFE's extension to longer text. We address each of your concerns below:
>
> ---
>
> ### **Q1: Clarification of Our Contributions.**
>
> > *"MAFE itself is not especially groundbreaking- many papers have developed fused representations using the concatenation or average of the image and text embeddings from CLIP. As such the main contribution is the extension to longer text."*
>
> **A1:** We clarify that our contributions extend beyond simple concatenation:
>
> 1. **MAFE Framework**: We propose a novel framework that fundamentally addresses CLIP's inherent limitations for VLM safety detection: (i) the 77-token constraint that prevents processing lengthy malicious prompts, and (ii) the inability to simultaneously process cross-modal information. Our progressive text aggregation with similarity-weighted pooling and cross-modal fusion enables CLIP to handle complex multimodal scenarios.
>
> 2. **Novel Discovery**: We are the first to systematically combine CLIP's [EOS] and [CLS] tokens and discover their distinct distributional patterns between benign and malicious prompts in the VLM jailbreak context (***Figure-3, Table-1***). This empirical finding provides new insights into multimodal safety detection.
>
> 3. **Lightweight VLMShield Detector**: Building on these foundations, we design an efficient and robust safety detector that achieves exceptional performance (≤2.13% OOD ASR, 96.33-100% benign accuracy) with minimal overhead (0.34s), significantly outperforming existing methods.
>
> We hope these clarifications demonstrate that our work establishes a principled and practical defense framework, and we respectfully request reconsideration of the contribution assessment.
>
> ---
>
> ### **Q2: Dataset Distributional Separation Analysis.**
>
> > - *"The empirical analysis raises suspicions that there are some tell-tale features distinguishing datasets and deeper investigation is needed to rule out the possibility the images and text are simply sampled from different distributions, making discrimination trivial. The too-good-to-be-true results in Table 3 are a smoking gun, where VLMShield is outperfoming SOTA methods by a huge margin. For example, ..."*
> > - *"Figure 3 shows not only clear separation between benign and malicious but also clear separation between the dataset sources, suggesting there may be other structural reasons why these datasets are well-separated. This sort of too-good-to-be-true result should be treated with suspicion..."*
>
> **A2:** Please see **[General Response #1—Dataset Distributional Separation Analysis](https://openreview.net/forum?id=pen4NG41j6&noteId=a4P14KmtsD)**.
>
> ---
>
> ### **Q3: Analysis of Figure-6~8 in Appendix-A.**
>
> > *"Selective mixing of some datasets in Fig 6/7/8 also point to some structural giveaways- it would be worthwhile to better understand and explain why some of the datasets selectively collapse onto others under selective modality changes- eg image-based onto direct malicious in Figs 7 and 8."*
>
> **A3:** The observed clustering patterns in ***Figure-6~8*** validate why complete multimodal understanding is essential:
>
> - ***Figure-6*** (Without Long Text Processing): Removing Progressive Text Aggregation causes truncation of texts beyond 75 tokens. Since harmful content often appears in the latter portions of prompts (e.g., jailbreak trigger instructions), truncation eliminates these malicious features, causing truncated malicious samples to drift toward benign clusters. This demonstrates the necessity of our long text processing mechanism.
>
> - ***Figure-7*** (Text-Only Features): Text-based jailbreaks intentionally craft prompts to appear benign in textual semantics—this is precisely their attack strategy. Similarly, image-based jailbreaks pair malicious images with innocent text (e.g., "Describe this image"). Without visual information, both attack types successfully masquerade as benign in text space, causing observed clustering with benign samples.
>
> - ***Figure-8*** (Image-Only Features): Image-based jailbreaks use adversarial perturbations or embedded instructions, manipulating visual features to appear safer, creating partial overlap in feature space. However, they remain distinguishable from truly benign images, explaining their intermediate clustering position.
>
> ***Figure-6~8*** collectively demonstrate that effective multimodal attack detection requires complete cross-modal semantic understanding. Attackers deliberately exploit modality gaps—appearing harmless in one modality while embedding malicious content in another. The modality-dependent clustering validates that MAFE's superior separation stems from capturing complementary multimodal semantics unavailable to single-modality analysis, confirming that cross-modal fusion is essential for robust defense.

---

> > ### Author Response · Authors · 2025-11-21
> > **Response to Reviewer-qPm8 Q4~Q8**
> >
> > ### **Q4: Future Attacks.**
> >
> > > *"The classification model is dependent on a corpus of labeled images which in an adversarial setting may open the door to novel attack methods that require manual detection and retraining."*
> >
> > **A4:** We appreciate this concern about generalization to future attacks.
> > - Our evaluation includes in-domain, out-of-domain and adaptive attacks, demonstrating strong robustness and generalization to unseen attack methods. These results reflect VLMShield's potential to handle novel attacks without immediate retraining.
> > - Moreover, we acknowledge that adversarial defense is inherently an evolving arms race where attackers and defenders continuously adapt. We hope our work contributes to advancing more robust defenses in this ongoing process.
> >
> > ---
> >
> > ### **Q5: Text Segmentation Strategy.**
> >
> > > *"How does MAFE handle text segmentation where, say, a single sentence runs for more than 75 tokens? How do you achieve 'semantic completeness and coherence'?"*
> >
> > **A5:** We employ overlapping chunks with 10-token overlaps (as stated in ***Line-178*** and ***Line-309***) to preserve semantic completeness and coherence across chunk boundaries. This design ensures that sentences spanning chunk boundaries remain contextually connected. Our experimental results demonstrate that this approach effectively maintains semantic integrity for classification purposes. We also include ablation studies on overlap size in the appendix (***Appendix-D.2***) to further validate this design choice.
> >
> > ---
> >
> > ### **Q6: Ablation Study.**
> >
> > > *"The kernel-density representation of the input text (Eq 4) seems like a good choice- did you perform any ablations over alternatives (eg the average of the embeddings)?"*
> >
> > **A6:** Please see **[General Response #3—Ablation Study](https://openreview.net/forum?id=pen4NG41j6&noteId=Nx6Mc0FzHO)**.
> >
> > ---
> >
> > ### **Q7: Single-Modality Input Handling.**
> >
> > > *"'this framework can handle multimodal and single-modality inputs seamlessly' - you don't describe how you handle single-modal inputs- do you just use a zero vector for the missing modality?"*
> >
> > **A7:** Yes. For single-modality inputs, we use a zero vector for the missing modality.
> >
> > ---
> >
> > ### **Q8: Presentation Issue.**
> >
> > > *"Formatting: margins between images and text are compressed making it difficult to discriminate between image captions and text body. These practices risk desk rejection."*
> >
> > **A8:** We apologize for the presentation issue. We have carefully optimized the layout in the revision to avoid this issue.

---

> > > ### Comment · Reviewer_qPm8 · 2025-11-25
> > >
> > > Thank you for your thorough rebuttal specifically addressing my questions and the questions from other reviewers. Although you have conducted some valuable investigation into the datasets, I remain somewhat skeptical of how easily the classes are separated using this approach. Nonetheless I have slightly boosted my soundness, contribution, and overall rating scores.

---

> > > > ### Author Response · Authors · 2025-11-26
> > > > **Response to Reviewer qPm8's Follow-up Comment**
> > > >
> > > > We sincerely thank the reviewer for the thoughtful feedback and the increased ratings. We appreciate your concern regarding the separability of classes using our MAFE approach and would like to provide additional mechanistic insights to address this important question.
> > > >
> > > > We have added analysis in ***Appendix-G***, examining why MAFE achieves effective class separation from an attention mechanism perspective. Through qualitative and quantitative analysis of attention patterns in CLIP's text and vision encoders (***Figure-14~15***, ***Table-18~19***), we demonstrate:
> > > >
> > > > **1. Semantic Aggregation Property.**
> > > >
> > > > Both [EOS] and [CLS] tokens function as semantic aggregators through the transformer's self-attention mechanism. Our analysis reveals that these tokens consistently consolidate the most discriminative information from their respective modalities through learned attention patterns. The 100% Top-1 aggregator ratios (***Table-18~19***) provide quantitative evidence that [EOS] and [CLS] reliably capture semantically central content—the core features that distinguish malicious from benign content.
> > > >
> > > > **2. Complementary Multimodal Information Capture.**
> > > >
> > > > The text [EOS] token captures semantic intent and linguistic patterns indicative of malicious queries (e.g., jailbreak trigger phrases, harmful instructions), while the visual [CLS] token captures visual anomalies characteristic of adversarial attacks (e.g., embedded harmful content, adversarial perturbations). Since multimodal attacks manifest through one or both of these channels, concatenating these representations enables MAFE to simultaneously monitor both attack vectors. This cross-modal complementarity explains why MAFE achieves superior separation compared to single-modality approaches—malicious prompts that may appear benign in one modality reveal their true nature when both modalities are jointly analyzed.
> > > >
> > > > These mechanistic insights demonstrate that MAFE's separability stems from leveraging CLIP's inherent semantic aggregation capabilities across both modalities, enabling comprehensive capture of multimodal attack characteristics. We hope this additional analysis addresses your concern about the ease of class separation and further validates our approach's effectiveness.

---

### Official Review · Reviewer_SYBb · 2025-10-31

**Soundness:** 2
**Presentation:** 3
**Contribution:** 2
**Rating:** 2
**Confidence:** 4

**Summary:**

The paper proposes VLMShield, a safety detector for multimodal jailbreak attacks. It relies on CLIP with MAFE, which can aggregate and extract longer text and concatenates text and image embeddings and classify the prompts as benign or harmful. Experiments show high in-domain and out-of-domain robustness and low latency when compared to existing defenses.

**Strengths:**

1. The approach is model-agnostic and does not require modifying or re-training the underlying VLM, which is useful in real deployment settings.

2. The detector is lightweight, making it suitable for real-time or production use.

3. The external structure also does not affect the downstream model performance

**Weaknesses:**

1. Motivation for using CLIP instead of the LVLM itself is unconvincing.
The authors argue that the goal is to build a unified detector that can process both text and image inputs, and CLIP is used because it offers modality-specific encoders and is extended via MAFE to handle long text. However, LVLMs already jointly process image and text inputs and naturally support long context. Therefore, the same effect could be achieved by performing classification directly within the LVLM, without introducing an external CLIP-based detector.
Additionally, recent work has shown that LVLM robustness can be improved directly, or that safety classification can be performed using intermediate hidden states within the LVLM itself, without requiring external encoders. See [1–4].

2. The claim that “existing defenses cannot efficiently handle multimodal inputs” is overstated.
Many LVLM-based safety moderation pipelines already support multi-image, multi-turn, and long-context moderation with modest overhead. Harmful intent can often be detected in the first few decoding steps, so the argument that LVLM-based detection is inherently inefficient is not well supported.

3. Distributional separation experiments may be misleading due to dataset bias.
Each malicious category corresponds to a different dataset, so the observed separation in t-SNE/MMD may reflect dataset-level differences rather than semantic differences in attack style. To support the claim that MAFE captures modality-specific attack patterns, the authors should evaluate clustering consistency across multiple datasets within the same attack category. Without that, the conclusion about capturing attack-type features is not justified.

4. Benign dataset coverage is weak, and the reported benign accuracy appears unrealistically high.
Only two out-of-domain benign datasets are used, and both are caption-style tasks, which are structurally simpler than real conversational or instruction-based benign prompts. More diverse benign data should be included to assess false positive behavior.
Additionally, the reported benign accuracy is unusually high. For example, the official Qwen2.5-VL-7B performance is around 67.1% on MM-Vet and 82.6% on MMBench, yet the paper reports nearly 100% accuracy. It is unclear whether standard splits and evaluation protocols were used. The paper should also report the performance of the unmodified Qwen2.5-VL model on the same test set to verify consistency. Clarification is needed to rule out evaluation mismatch or data leakage.

References
[1] Renjie Pi et al., "MLLM-Protector: Ensuring MLLM’s Safety Without Hurting Performance", arXiv:2401.02906, 2024.
[2] Zhendong Liu et al., "Enhancing Vision-Language Model Safety Through Progressive Concept-Bottleneck-Driven Alignment", arXiv:2411.11543, 2024.
[3] Wenhan Yang et al., "Bootstrapping LLM Robustness for VLM Safety via Reducing the Pretraining Modality Gap", arXiv:2505.24208, 2025.
[4] Zhenhong Zhou et al., "How Alignment and Jailbreak Work: Explain LLM Safety through Intermediate Hidden States", arXiv:2406.05644, 2024.

**Questions:**

1. Why is CLIP preferred over simply using the VLM’s own joint embedding space for classification? Did you try a simple linear classifier over the VLM’s pooled embeddings?
2. How do you control for dataset-induced feature separation in your MMD experiments?
3. How well does the detector handle benign prompts requiring complex reasoning, rather than descriptive captioning tasks?

---

> ### Author Response · Authors · 2025-11-21
> **Response to Reviewer-SYBb Q1**
>
> We sincerely thank Reviewer-SYBb for the constructive feedback and for recognizing our model-agnostic design, lightweight architecture, and preservation of downstream model performance. We address each of your concerns below:
>
>
> ### **Q1: Clarification of the Motivation.**
>
> > - *"Motivation for using CLIP instead of the LVLM itself is unconvincing... LVLMs already jointly process image and text inputs and naturally support long context... recent work has shown that LVLM robustness can be improved directly, or that safety classification can be performed using intermediate hidden states within the LVLM itself [1–4]."*
> > - *"The claim that 'existing defenses cannot efficiently handle multimodal inputs' is overstated. Many LVLM-based safety moderation pipelines already support multi-image, multi-turn, and long-context moderation with modest overhead..."*
>
> **A1:** We clarify that the referenced works [1–4] actually fall within the defense taxonomy we established in ***Section-2.2***, where we categorize existing defenses into internal and external approaches. Specifically:
>
> - **[2, 3, 4] are internal defenses** that require white-box access to VLM internals: [2] introduces concept bottlenecks requiring model retraining, [3] applies regularization during pretraining to reduce modality gaps, and [4] trains a linear classifier on intermediate hidden states of LLMs. All require access to model parameters and internal representations.
>
> - **[1] is an external defense**: [1] trains a harm detector on VLM outputs similar to ECSO in our experiments, operating independently of model internals.
>
> Our motivation, stated in ***Line-57~58***, is to achieve **efficient and robust defense**. The referenced methods align with the internal and external defense categories we discussed in ***Section-2.2***, and thus share the same limitations we identified for these approaches:
>
> - **Regarding Efficiency:**
>   - Methods [2, 3] require model retraining/fine-tuning with substantial computational overhead, making them impractical for deployment scenarios where models cannot be modified.
>   - Method [4], while detecting harmful intent through internal hidden states, still requires VLM forward passes through billions of parameters to access these representations. For large VLMs (e.g., LLaVA-1.5-13B with 13B parameters), this computational burden is substantial, and the high-dimensional embeddings further increase processing overhead.
>   - Method [1] requires complete VLM output generation before detection (similar to ECSO in ***Table-5***), resulting in 2.52s detection overhead compared to our 0.34s.
>
> - **Regarding Robustness:**
>   - Methods [2, 3, 4] are model-dependent—their learned safety features vary across different VLM architectures. As models change, defense effectiveness changes accordingly, limiting generalizability.
>   - Method [1] relies on output detection similar to ECSO. Tables 2-3 demonstrate that output-monitoring approaches achieve 18.39-43.06% ASR, significantly worse than our ≤2.13% ASR, indicating insufficient robustness.
>
> - In contrast, VLMShield leverages CLIP, a component present in virtually all VLMs, ensuring consistent protection across models without retraining (***Table-2~5***).
>
> To directly address your suggestion, we have conducted additional experiments in ***Appendix-A.3***, comparing VLMShield against using VLM internal hidden state representations (the approach analogous to [4]). Specifically, we extract embeddings from the last hidden layer of VLMs and visualize their distributions across different attack categories. Results confirm:
>
> - **Robustness**: Internal hidden state representations show poor discriminative ability across attack categories, with overlapping distributions between benign and malicious samples. This demonstrates that model-dependent features fail to generalize well across diverse attack types, resulting in weak robustness.
>
> - **Efficiency**: Extracting internal representations requires forward passes through billions of VLM parameters, and the resulting high-dimensional embeddings incur substantial processing overhead. This results in detection times far exceeding our 0.34s, making real-time deployment impractical.
>
> These results validate our design choice: **VLMShield achieves both superior efficiency and robustness** by operating on CLIP features at the input level, fulfilling our stated motivation in a way that existing methods cannot.
>
> Reference:
>
> [1] Renjie Pi et al., "MLLM-Protector: Ensuring MLLM’s Safety Without Hurting Performance", arXiv:2401.02906, 2024.
>
> [2] Zhendong Liu et al., "Enhancing Vision-Language Model Safety Through Progressive Concept-Bottleneck-Driven Alignment", arXiv:2411.11543, 2024.
>
> [3] Wenhan Yang et al., "Bootstrapping LLM Robustness for VLM Safety via Reducing the Pretraining Modality Gap", arXiv:2505.24208, 2025.
>
> [4] Zhenhong Zhou et al., "How Alignment and Jailbreak Work: Explain LLM Safety through Intermediate Hidden States", arXiv:2406.05644, 2024.

---

> > ### Author Response · Authors · 2025-11-21
> > **Response to Reviewer-SYBb Q2~Q4**
> >
> > ### **Q2: Dataset Distributional Separation Analysis.**
> >
> > > *"Distributional separation experiments may be misleading due to dataset bias. Each malicious category corresponds to a different dataset, so the observed separation in t-SNE/MMD may reflect dataset-level differences rather than semantic differences in attack style. To support the claim that MAFE captures modality-specific attack patterns, the authors should evaluate clustering consistency across multiple datasets within the same attack category. Without that, the conclusion about capturing attack-type features is not justified"*
> >
> > **A2:** Please see **[General Response #1—Dataset Distributional Separation Analysis](https://openreview.net/forum?id=pen4NG41j6&noteId=a4P14KmtsD)**.
> >
> >
> > ---
> >
> > ### **Q3: Clarification of the Benign Evaluation.**
> >
> > > *"Additionally, the reported benign accuracy is unusually high. For example, the official Qwen2.5-VL-7B performance is around 67.1% on MM-Vet and 82.6% on MMBench, yet the paper reports nearly 100% accuracy. It is unclear whether standard splits and evaluation protocols were used. The paper should also report the performance of the unmodified Qwen2.5-VL model on the same test set to verify consistency. Clarification is needed to rule out evaluation mismatch or data leakage."*
> >
> > **A3:** We appreciate the opportunity to clarify this confusion.
> > - Our reported "benign accuracy" measures **classification accuracy**—the percentage of benign prompts correctly identified as benign by VLMShield—not the VLM's task performance. Since VLMShield operates at the input level without modifying prompts, it does not affect downstream task performance.
> > - The 67.1% MM-Vet and 82.6% MMBench scores you mentioned reflect the VLM's reasoning capabilities, while our near-100% accuracy indicates that VLMShield rarely misclassifies benign prompts as malicious (low false positive rate).
> > - This distinction is explicitly defined in ***Section-5 (Metrics)***.
> >
> > ---
> >
> > ### **Q4: More Results on Benign Datasets.**
> >
> > > - *"Benign dataset coverage is weak, and the reported benign accuracy appears unrealistically high. Only two out-of-domain benign datasets are used, and both are caption-style tasks, which are structurally simpler than real conversational or instruction-based benign prompts."*
> > > - *"How well does the detector handle benign prompts requiring complex reasoning, rather than descriptive captioning tasks?"*
> >
> > **A4:** We additionally evaluated VLMShield on VQAv2, a visual question answering benchmark requiring complex reasoning beyond simple captioning, where it achieves 99.38% benign accuracy.

---

> > > ### Author Response · Authors · 2025-11-26
> > > **Follow-up on Rebuttal Response**
> > >
> > > Dear Reviewer SYBb,
> > >
> > > We hope this message finds you well. We sincerely appreciate the time and effort you devoted to reviewing our manuscript and providing valuable feedback.
> > >
> > > We submitted our detailed rebuttal addressing all the concerns you raised and have made substantial revisions to the manuscript, including additional experiments and analyses as suggested. **As the discussion phase is approaching its conclusion**, we wanted to respectfully follow up to see if you had the opportunity to review our response.
> > >
> > > We remain committed to addressing any remaining questions or concerns you may have. If there are any aspects of our rebuttal that require further clarification, we would be grateful for the opportunity to address them during the remaining discussion period.
> > >
> > > Thank you again for your valuable contribution to improving our work. We look forward to hearing from you.
> > >
> > > Best regards,
> > >
> > > The Authors

---

### Official Review · Reviewer_cXjt · 2025-11-02

**Soundness:** 3
**Presentation:** 3
**Contribution:** 3
**Rating:** 6
**Confidence:** 3

**Summary:**

This paper introduces VLMShield, a lightweight and black-box defense for multimodal large models. The method first builds a Multimodal Aggregated Feature Extractor (MAFE) based on CLIP to handle long text inputs by segmenting and aggregating them through [EOS] similarity weighting, then fuses the text and image representations ([EOS] and [CLS]) into a 1536-dim feature. A small MLP classifier is trained to distinguish benign from malicious inputs. The approach serves as a plug-and-play front-end filter before any VLM. Experiments across multiple jailbreak scenarios demonstrate low attack success rates and negligible latency, with good generalization to unseen datasets. The authors also evaluate adaptive attacks and release anonymized code for reproduction.

**Strengths:**

1.The motivation is clear and realistic. The paper focuses on a deployable, black-box input-level defense that fits real-world multimodal systems.

2.The design is simple but effective. CLIP-based long-text aggregation and multimodal fusion generate separable features, and a tiny MLP achieves strong detection accuracy with minimal cost.

3.The evaluation is broad and convincing. The authors include IND/OOD datasets, benign accuracy, efficiency, and adaptive attacks, comparing against several representative baselines such as ASTRA, VLMGuard, JailGuard, and MirrorCheck.

4.Results are impressive: the defense achieves very low ASR (<2% in most cases) and maintains almost full benign accuracy with <10% latency overhead.

5.Training details, hyperparameters, and code are clearly provided for reproducibility.

**Weaknesses:**

1.The methodological novelty is limited. The approach mainly combines CLIP feature aggregation and a small classifier—solid engineering, but conceptually incremental.

2.The adaptive threat model is relatively weak. Stronger surrogate or gradient-based attacks directly targeting the MAFE layer are not explored, so the robustness might be overestimated.

3.The evaluation lacks coverage on larger or closed-source VLMs, and does not report ROC or AUROC curves to support threshold selection and deployment tuning.

**Questions:**

1.How sensitive is the method to design choices such as chunk size, similarity weighting, and the choice of CLIP backbone (e.g., RN50, ViT-H, SigLIP)? Please provide a more complete sensitivity or ablation analysis.

2.How would VLMShield perform under stronger adaptive attacks that jointly optimize over MAFE representations or combine layout/typographic perturbations? Can the authors discuss or experiment on this scenario?

---

> ### Author Response · Authors · 2025-11-21
> **Response to Reviewer-cXjt**
>
> We sincerely thank Reviewer-cXjt for the thoughtful review and for recognizing our clear motivation, effective design, comprehensive evaluation, and reproducibility. We address each of your concerns below:
>
> ### **Q1: Clarification of Our Contributions**.
> > "*The methodological novelty is limited. The approach mainly combines CLIP feature aggregation and a small classifier—solid engineering, but conceptually incremental.*"
>
> **A1:** We clarify that our contributions extend beyond the engineering:
>
> 1. **MAFE Framework**: We propose a novel framework that fundamentally addresses CLIP's inherent limitations for VLM safety detection: (i) the 77-token constraint that prevents processing lengthy malicious prompts, and (ii) the inability to simultaneously process cross-modal information. Our progressive text aggregation with similarity-weighted pooling and cross-modal fusion enables CLIP to handle complex multimodal scenarios.
>
> 2. **Novel Discovery**: We are the first to systematically combine CLIP's [EOS] and [CLS] tokens and discover their distinct distributional patterns between benign and malicious prompts in the VLM jailbreak context (***Figure-3***, ***Table-1***). This empirical finding provides new insights into multimodal safety detection.
>
> 3. **Lightweight VLMShield Detector**: Building on these foundations, we design an efficient and robust safety detector that achieves exceptional performance (≤2.13% OOD ASR, 96.33-100% benign accuracy) with minimal overhead (0.34s), significantly outperforming existing methods.
>
> These contributions collectively establish a principled and practical defense framework for securing multimodal AI systems.
>
> ---
>  ### **Q2: Adaptive Attacks**.
> > *"The adaptive threat model is relatively weak. Stronger surrogate or gradient-based attacks directly targeting the MAFE layer are not explored, so the robustness might be overestimated."*
>
> **A2:** Please see **[General Response #2—Adaptive Attacks](https://openreview.net/forum?id=pen4NG41j6&noteId=K9DSw3xIWI)**.
>
> ---
>  ### **Q3: Clarification of VLM Selection**.
> > *"The evaluation lacks coverage on larger or closed-source VLMs."*
>
> **A3:** We appreciate the opportunity to clarify this confusion.
> - A key advantage of VLMShield is its **model-independent** design, unlike some defense methods whose effectiveness varies across different models (eg. VLMGuard, ASTRA, etc.).
> - Since our defense operates purely on CLIP-extracted input features without accessing VLM internals, it achieves consistent detection performance regardless of the target model's size or accessibility.
> - As shown in ***Table-2~4***, VLMShield produces identical results across different architectures and directly transfers to larger or closed-source VLMs without retraining.
>
> ---
> ### **Q4: Ablation Study.**
>
> > - *"...does not report ROC or AUROC curves to support threshold selection and deployment tuning."*
> > - *"How sensitive is the method to design choices such as chunk size, similarity weighting, and the choice of CLIP backbone?"*
>
> **A4:** Please see **[General Response #3—Ablation Study](https://openreview.net/forum?id=pen4NG41j6&noteId=Nx6Mc0FzHO)**.

---

> > ### Comment · Reviewer_cXjt · 2025-11-23
> >
> > Thank you for the author's response. I am glad to see that the author has added many experiments, which has improved the completeness of the current article. Although the method level is similar to traditional Bert+MLP for classification to identify multimodal malicious attacks, this lightweight tool is efficient and effective. Overall, I am still willing to maintain the current positive score.

---

> > > ### Author Response · Authors · 2025-11-24
> > > **Thank you so much for your positive feedback!**
> > >
> > > Dear Reviewer cXjt,
> > >
> > > Thank you so much for your positive feedback on our response and the revised manuscript. We are grateful for your recognition that we have improved the completeness of the article. We have worked diligently to address all concerns raised during the review process. If there are any remaining issues or aspects that need further clarification, we would be more than happy to address them promptly.
> > >
> > > Given your acknowledgment of these improvements, we would like to respectfully inquire whether you might consider adjusting the score to better reflect the enhanced quality of the revised work. We greatly appreciate your time and expertise throughout this process.
> > >
> > > Best regards,
> > >
> > > The Authors

---

### Author Response · Authors · 2025-11-21
**[3/3] General Response #3—Ablation Study (R-cXjt, R-qPm8 & R-8s2m)**

We thank all reviewers for these valuable suggestions. We have conducted comprehensive ablation studies in ***Section-6 (RQ5)*** and ***Appendix-D***, covering five key aspects:

### **1. Chunk Size.**

The choice of 75-token chunks is motivated by CLIP's architectural constraints: CLIP processes sequences of 77 tokens, with 2 positions reserved for special tokens ([SOS] and [EOS]), leaving 75 positions for actual content. This design maximally utilizes CLIP's token capacity.

We evaluated performance on MM-Vet (benign) and text_based_jailbreak_28K (malicious) datasets:

| Chunk Size | Overlap | Benign ACC(%) | ASR(%) | Detection Time(s) |
|------------|---------|---------------|--------|-------------------|
| 50 | 10 | 96.33 | 0.00 | 0.37 |
| 75 | 10 | 96.33 | 0.00 | 0.34 |

**Analysis:** Chunk size variations show minimal impact on detection effectiveness but directly affect computational efficiency. The 75-token configuration achieves the same high accuracy while reducing detection time (0.34s vs 0.37s). Therefore, we select CLIP's maximum acceptable length (75 tokens) to minimize the number of chunks and maximize detection efficiency.

### **2. Overlap Size.**

We compared different overlap sizes (0, 5, 10, 20 tokens) on the same datasets:

| Chunk Size | Overlap | Benign ACC(%) | ASR(%) | Detection Time(s) |
|------------|---------|---------------|--------|-------------------|
| 75 | 0 | 96.28 | 0.47 | 0.30 |
| 75 | 5 | 96.30 | 0.36 | 0.33 |
| 75 | 10 | 96.33 | 0.00 | 0.34 |
| 75 | 20 | 96.36 | 0.00 | 0.47 |

**Analysis:** The 10-token overlap provides optimal balance: maintaining semantic continuity across chunk boundaries while avoiding excessive computational overhead (0.47s vs 0.34s for 20-token overlap).

### **3. Text Aggregation Method.**

We compared three aggregation strategies on MM-Vet (benign) and text_based_jailbreak_28K (malicious):

| Aggregation Method | Benign ACC(%) | ASR(%) | MMD |
|-------------------|---------------|--------|-----|
| Simple Average | 96.30 | 1.46 | 0.692 |
| Similarity-weighted (Ours) | 96.33 | 0.00 | 0.835 |
| MAX-Pooling | 94.29 | 5.39 | 0.507 |

**Analysis:** Simple averaging treats all chunks equally, potentially introducing harmful content into benign representations. MAX-Pooling captures extreme features but loses overall context. Our similarity-weighted approach automatically emphasizes semantically central content (malicious or benign), achieving superior separability (higher MMD) and lower ASR while reducing the need for manual intervention.

### **4. CLIP Backbone.**

We evaluated different CLIP architectures on MM-Vet and text_based_jailbreak_28K, assessing the impact of backbone choice beyond ResNet-based alternatives (which showed inferior performance in dataset bias experiments):

| CLIP Backbone | Benign ACC(%) | ASR(%) | Detection Time(s) |
|--------------|---------------|--------|-------------------|
| ViT-L/14 | 96.33 | 0.00 | 0.34 |
| ViT-H/14 | 97.04 | 0.00 | 0.57 |
| SigLIP-L | 95.17 | 3.05 | 0.35 |

**Analysis:** ViT-H/14 offers slightly better performance but increased computational cost. ViT-L/14 provides the best balance for safety detection. SigLIP-L shows reduced performance, possibly due to its sigmoid-based training diverging from CLIP's contrastive learning approach.

### **5. Detection Threshold.**

We evaluated performance across different classification thresholds on MM-Vet (benign) and MM-SafetyBench (malicious):

| Threshold | Benign ACC(%) | ASR(%) |
|-----------|---------------|--------|
| 0.3 | 100.00 | 10.04 |
| 0.4 | 99.34 | 5.27 |
| 0.5 | 96.33 | 0.00 |
| 0.6 | 90.46 | 0.00 |
| 0.7 | 83.84 | 0.00 |

**Analysis:** Our default threshold of 0.5 achieves optimal balance, reaching 96.33% benign accuracy with 0.00% ASR. Lower thresholds (0.3-0.4) achieve higher benign accuracy (99.34-100%) but allow more attacks (ASR 5.27-10.04%). Higher thresholds (0.6-0.7) maintain perfect attack blocking (ASR 0.00%) but sacrifice benign accuracy (83.84-90.46%). The threshold validates our design's rationality, balancing attack success prevention with high benign accuracy. For deployment, practitioners can adjust thresholds based on specific requirements: security-critical applications may prefer ≥0.6 (accepting lower benign accuracy), while our default 0.5 provides the best overall trade-off.

---

### Author Response · Authors · 2025-11-21
**[2/3] General Response #2—Adaptive Attacks (R-cXjt, R-vPvD & R-8s2m)**

We sincerely thank all reviewers for their concerns about adaptive robustness. We address these concerns from two perspectives:

### **1. Clarification of Our Original Adaptive Attack Method.**

As described in ***Line-481~504***, our original adaptive attacks optimize text or image inputs to alter MAFE representations for evading VLMShield detection, rather than directly modifying MAFE features. This is because directly manipulating MAFE representations cannot be reverse-engineered back into valid text or image inputs that VLMs can accept. Specifically, [EOS] and [CLS] tokens undergo complex transformations through multiple Transformer layers before MAFE aggregation, and these intermediate computational steps are not directly invertible for meaningful input recovery. Therefore, MAFE representations are fundamentally resistant to reverse-engineering, making direct feature manipulation infeasible—attackers must instead optimize actual text or image inputs to influence the resulting MAFE representations.


### **2. Additional Adaptive Attacks.**

**2.1 Combined Perturbation Attacks.**

We implement two types of combined attacks that simultaneously exploit multiple attack vectors:

**(1) Joint Text+Image Optimization.**

We simultaneously apply GCG optimization (targeting text EOS embeddings) and HADES optimization (targeting image CLS embeddings) to jointly manipulate the concatenated MAFE representations:

$$L_{joint} = (1 - \lambda) \cdot (L_{adv}^{text} + L_{adv}^{image}) + \lambda \cdot L_{evade}^{joint}$$

where the loss jointly optimizes both text and image perturbations to evade MAFE-based detection while maximizing harmful output generation.

- **Results:**

| Attack Type | Strategy | ASR (%) ↓ | HGR (%) ↓ | Effective ASR (%) ↓ |
|-------------|----------|-------|-------|-----------------|
| **Joint Text+Image Optimization** | λ=0 | 1.63 | 100 | 1.63 |
| | λ=0.5 | 4.83 | 42.74 | 2.06 |
| | λ=1 | 6.02 | 12.37 | 0.74 |

**(2) Multi-Perturbation Image Attacks.**

We combine multiple image-based attack strategies targeting the image modality: combining FigStep with HADES adversarial perturbations.

- **Results:**

| Attack Type | Strategy | ASR (%) ↓ | HGR (%) ↓ | Effective ASR (%) ↓ |
|-------------|----------|-------|-------|-----------------|
| **Multi-Perturbation Image** | λ=0 | 1.25 | 100 | 1.25 |
| | λ=0.5 | 4.06 | 45.02 | 1.83 |
| | λ=1 | 5.63 | 14.04 | 0.79 |

These results demonstrate that even under stronger combined adaptive attacks, VLMShield maintains low effective ASR, validating the fundamental insight that **successfully evading VLMShield reduces the harmfulness of generated content, resulting in low effective ASR.**


**2.2 Dilution Attacks.**

We implement a new attack strategy that directly exploits text chunking by embedding minimal malicious content within extensive benign context, systematically varying the length disparity.

- **Experimental Design:**  Using AdvBench-M prompts, we created scenarios with varying chunk counts (5, 10, 20, 50, 100 chunks), where malicious content occupies only 1 chunk, with remaining chunks filled with benign content from GPT4V-Caption dataset.

- **Results:**

| Total Chunks | Malicious Chunks | ASR (%) | HGR (%) | Effective ASR (%) |
|--------------|------------------|---------|---------|-------------------|
| 5 | 1 | 0.48 | 100 | 0.48 |
| 10 | 1 | 0.83 | 94.2 | 0.78 |
| 20 | 1 | 1.12 | 78.5 | 0.90 |
| 50 | 1 | 4.97 | 62.7 | 3.12 |
| 100 | 1 | 8.73 | 43.8 | 3.82 |

Even with 100 chunks, VLMShield maintains stable performance with effective ASR < 4%. We also tested variations in malicious content placement (beginning/middle/end) and malicious content distribution across multiple chunks—detection effectiveness remained consistently high.

The fundamental reason lies in CLIP's embedding space characteristics. Malicious intent inherently represents core semantic content, which naturally receives higher similarity weights during MAFE aggregation, displaying better centrality even under dilution. As dilution increases, harmful generation rate dramatically drops (100% → 43.8%), because excessive benign context confuses the downstream VLM, creating an inherent trade-off that fundamentally limits attack effectiveness.

Current results demonstrate that **VLMShield effectively resists sophisticated dilution-based adaptive attacks with effective ASR below 4%**, maintaining practical security. Future work could further reduce the ASR through enhanced mechanisms such as anomaly detection and maximum pooling strategies.

### **3. Conclusion.**

Through the evaluation of multiple adaptive attack strategies, we demonstrate that VLMShield maintains robust defense with maximum effective ASR of 3.82%. The fundamental trade-off between evasion success and attack effectiveness, validates VLMShield's robustness against sophisticated adaptive adversaries with full knowledge of our defense mechanism.

---

### Author Response · Authors · 2025-11-21
**[1/3] General Response #1—Dataset Distributional Separation Analysis (R-SYBb & R-qPm8)**

We thank all reviewers for raising this concern about potential dataset-level artifacts. We clarify that **each dataset** in our experiments contains samples with **diverse characteristics**: images vary widely in resolution, aspect ratios, and content types, while text prompts range from short queries to lengthy instructions with diverse linguistic patterns. This natural diversity within each dataset makes trivial discrimination based on superficial features (e.g., uniform image size) unlikely.

To rigorously validate that MAFE captures genuine semantic attack patterns rather than dataset artifacts, we have conducted extensive additional analysis in ***Line-250~260*** and ***Appendix-A.3*** (***Page-15~17***). Specifically, we compare three feature extraction approaches across different dataset configurations:

### 1. **Feature Extraction Methods.**

- Traditional Feature Extraction: ResNet for image features + TF-IDF for text features, concatenated.
- VLM Internal Representations: Embeddings from the last hidden layer of VLMs (Qwen2.5-VL-7B-Instruct).
- Our Proposed MAFE: CLIP-based multimodal aggregated features.


### 2. **Experimental Configurations.**

-  Cross-Category Datasets: We select one representative dataset from each of the three malicious categories (Image-based Jailbreak, Text-based Jailbreak, Direct Malicious) and visualize their feature distributions against benign datasets.

-  Within-Category Datasets: For each malicious category, we separately evaluate multiple datasets from the same category against benign datasets to test clustering consistency.


### 3. **Results.**

**(1) Cross-Category Analysis (Figure-9, Appendix-A.3.1).**
- Traditional features and VLM internal representations fail to achieve clear separation between benign and malicious samples.
- Only MAFE successfully separates malicious from benign samples while maintaining consistent patterns across different datasets within the same attack category.

**(2) Within-Category Analysis (Figure-10~12, Appendix-A.3.2).**
- Across Image-based Jailbreak datasets (***Figure-10***), Text-based Jailbreak datasets (***Figure-11***), and Direct Malicious datasets (***Figure-12***), MAFE consistently maintains a clear benign-malicious separation.
- Different datasets within the same attack category converge in MAFE's feature space, where traditional and VLM-based features show inconsistent patterns across datasets from the same category.

### 4. **Conclusion.**
These experiments demonstrate that:
- MAFE successfully captures attack patterns that generalize across multiple datasets within the same category.
- Without MAFE processing, standard feature extraction methods fail to achieve meaningful separation, indicating that clear clustering is not an artifact of dataset-level differences.

We hope these extensive additional experiments convincingly address the concern about dataset bias and validate that MAFE's discriminative power stems from capturing genuine multimodal attack semantics.

---

### Author Response · Authors · 2025-12-01
**Summary of Reviewer Feedback and Request for Area Chair Consideration**

Dear Area Chairs,

Please find below a summary of the points raised by reviewers and how we responded to them.

### - **Scores (after rebuttal): 6,6,6,4,2 Confidence: 4,3,3,4,4**

One reviewer improved their score from 2→4 after our initial rebuttal, and we addressed their follow-up questions (no response received). The score of 2 reflects several misunderstandings  (all addressed in our response) with no engagement in the rebuttal phase.

---

### - **Response to Reviewer-cXjt**
In our response to Reviewer-cXjt, we have summarized the points we addressed:

- Clarification of methodological contributions (MAFE framework, novel distributional discovery, lightweight detector)
- Stronger adaptive attacks: joint text+image optimization, multi-perturbation attacks, dilution attacks
- Comprehensive ablation studies: chunk size, overlap, aggregation methods, CLIP backbone, detection thresholds
- VLM selection and model-independent design clarification

**Result:** Reviewer maintained a **positive score 6**, acknowledging improved completeness.

---
### - **Response to Reviewer-qPm8**
In our Initial response to Reviewer-qPm8, we have summarized the points we addressed:

- Clarified contributions (MAFE framework, novel distributional discovery)
- Dataset bias analysis: Cross-category and within-category experiments (***Appendix-A.3***, ***Figures-9~12***)
- Modality-specific clustering explanation (***Figures-6~8***)
- Text segmentation strategy and semantic coherence preservation via 10-token overlaps
- Single-modality input handling (zero vector for missing modality)
- Ablation on text aggregation methods (simple average vs similarity-weighted vs max-pooling)

**Result:** Reviewer qPm8 raised their score **from 2 to 4**, but raised concern about class separability.
In our follow-up response (https://openreview.net/forum?id=pen4NG41j6&noteId=7LhpqooHN9), we have summarized the points we addressed:

- Added mechanistic insights via attention analysis ***(Appendix-G, Figures-14~15***, ***Tables-18~19)***
- Demonstrated [EOS]/[CLS] tokens as semantic aggregators (100% Top-1 ratios)
- Explained complementary multimodal information capture

We believe our supplement has addressed the reviewers' new concerns, but as of writing, we have not received a response.

---
### - **Response to Reviewer-SYBb**

In our response to Reviewer-SYBb, we clarified **several misunderstandings of content already present in our original submission**:

- Motivation for CLIP vs VLM internal representations: Comparative experiments demonstrating efficiency and robustness advantages (***Appendix-A.3***).
- Defense taxonomy and positioning of related works [1-4]
- Dataset separation: Extensive analysis proving MAFE captures semantic patterns, not artifacts (***Figures-9~12***)
- Benign evaluation clarification: Added VQAv2 results (99.38% accuracy)

**Result:** Since we believe we have thoroughly addressed all issues and clarified the misunderstandings that led to the low rating 2, we requested a reassessment of their evaluation; as of this writing, we have not received a response.

---
### - **Response to Reviewer-vPvD**
In our response to Reviewer-vPvD, we have summarized the points we addressed:

- Adaptive attacks: Combined perturbation attacks and dilution attacks (maximum effective ASR 3.82%)
- MAFE reverse-engineering resistance explanation
- Failure case analysis: Qualitative analysis of false positives/negatives (***Appendix-F***)

**Result:** We have thoroughly addressed all concerns from this reviewer who initially gave a **positive rating 6**; as of this writing, we have not received a response to our rebuttal.

---
### - **Response to Reviewer-8s2m**
In our response to Reviewer-8s2m, we have summarized the points we addressed:

- Detection threshold ablation (0.3-0.7): Optimal balance at 0.5
- Overlapping chunk strategy clarification: 10-token overlap between consecutive 75-token chunks, with ablation showing optimal balance
- Multiple single-modality defenses cost comparison
- Output monitoring mechanism clarification: Using MLLM-based output checking (no regeneration), showing identical ASR to ECSO
- CIDER/MirrorCheck performance explanation
- Dilution attack robustness (effective ASR <4% at 100 chunks)

**Result:** We have thoroughly addressed all concerns from this reviewer who initially gave a **positive rating 6**; as of this writing, we have not received a response to our rebuttal.

---
### - **Summary**

We have thoroughly addressed all reviewer concerns with substantial additional experiments. ***Two reviewers acknowledged improvements, and three reviewers have not yet responded, but we believe their concerns are fully addressed.***

We hope that the area chair considers this context in their evaluation of the VLMShield paper.

Thank you!

The Authors

---

### Meta-Review · Area_Chair_WMgN · 2026-01-06

**Summary:**

In view of the fact that Vision-Language Models (VLMs) face significant safety vulnerabilities from malicious prompt attacks and that the existing defenses suffer from efficiency and robustness, the authors in this work claim to first propose the Multimodal Aggregated Feature Extraction (MAFE) framework that enables CLIP to handle long text and fuse multimodal information into unified representations.
After rebuttal, Reviewer cXjt commented that ``the method level is similar to traditional Bert+MLP for classification to identify multimodal malicious attacks, this lightweight tool is efficient and effective.'' and maintained the current positive score 6.
Reviewer qPm8 commented that ``(s)he remained somewhat skeptical of how easily the classes are separated using this approach.'' but did not further respond to the authors' response.
Reviewer vPvD raised the concerns of lack of analysis regarding adaptivity and defense circumvention and insufficient investigation of failure cases and false positives/negatives, but did not further respond to the authors' follow-up responses.
For comments from Reviewer-SYBb, the authors claimed that they clarified several misunderstandings of content already present in our original submission.
For comments from Reviewer 8s2m, the authors clarified some unclear parts.

Overall, the authors have responded to reviewers' comments and added several experimental results.
However, this paper needs remarkable revisions to achieve the standard of ICLR, and is not suggested to be accepted.

**Reviewer Concerns:**

Based on the Summary of Reviewer Feedback and Request for Area Chair and the rebuttals to reviewers' comments, the authors indeed have addressed the raised concerns.
However, the responses need the paper to be remarkably revised.

**Reviewer Scores:**

none

---

### Decision · Program_Chairs · 2026-01-26

Reject